# Adaptive Machine Unlearning

Varun Gupta[1], Christopher Jung[1], Seth Neel[2],
Aaron Roth[1], Saeed Sharifi-Malvajerdi[1], and Chris Waites[3]

[1]University of Pennsylvania
[2]Harvard University
[3]Stanford University

## Abstract

Data deletion algorithms aim to remove the influence of deleted data points from trained models at a cheaper computational cost than fully retraining those models. However, for sequences of deletions, most prior work in the non-convex setting gives valid guarantees only for sequences that are chosen *independently* of the models that are published. If people choose to delete their data as a function of the published models (because they don't like what the models reveal about them, for example), then the update sequence is *adaptive*. In this paper, we give a general reduction from deletion guarantees against adaptive sequences to deletion guarantees against non-adaptive sequences, using differential privacy and its connection to max information. Combined with ideas from prior work which give guarantees for non-adaptive deletion sequences, this leads to extremely flexible algorithms able to handle arbitrary model classes and training methodologies, giving strong provable deletion guarantees for adaptive deletion sequences. We show in theory how prior work for non-convex models fails against adaptive deletion sequences, and use this intuition to design a practical attack against the SISA algorithm of Bourtoule et al. [2021] on CIFAR-10, MNIST, Fashion-MNIST.

## 1 Introduction

Businesses like Facebook and Google depend on training sophisticated models on user data. Increasingly—in part because of regulations like the European Union's General Data Protection Act and the California Consumer Privacy Act—these organizations are receiving requests to delete the data of particular users. But what should that mean? It is straightforward to delete a customer's data from a database and stop using it to train *future* models. But what about models that have already been trained using an individual's data? These are not necessarily safe; it is known that individual training data can be exfiltrated from models trained in standard ways via *model inversion* attacks [Shokri et al., 2017, Veale et al., 2018, Fredrikson et al., 2015]. Regulators are still grappling with when a trained model should be considered to contain personal data of individuals in the training set and the potential legal implications. In 2020 draft guidance, the U.K.'s Information Commissioner's Office addressed how to comply with data deletion requests as they pertain to ML models:

> *If the request is for rectification or erasure of the data, this may not be possible without re-training the model...or deleting the model altogether* [ICO, 2020].

Fully retraining the model every time a deletion request is received can be prohibitive in terms of both time and money—especially for large models and frequent deletion requests. The problem of *data deletion* (also known as *machine unlearning*) is to find an algorithmic middle ground between the compliant but impractical baseline of retraining, and the potentially illegal standard of doing nothing. We iteratively update models as deletion requests come in, with the twin goals of having

35th Conference on Neural Information Processing Systems (NeurIPS 2021).

computational cost that is substantially less than the cost of full retraining, and the guarantee that the models we produce are (almost) indistinguishable from the models that would have resulted from full retraining.

After an initial model is deployed deletion requests arrive over time as users make decisions about whether to delete their data. It is easy to see how these decisions may be *adaptive* with respect to the models. For example, security researchers may publish a new model inversion attack that identifies a specific subset of people in the training data, thus leading to increased deletion requests for people in that subset. In this paper we give the first machine unlearning algorithms that both have rigorous deletion guarantees against these kind of adaptive deletion sequence, and can accommodate arbitrary non-convex models like deep neural networks without requiring pretraining on non-user data.

## 1.1 Main Results

The deletion guarantees proven for several prior methods crucially rely on the implicit assumption that the points that are deleted are independent of the randomness used to train the models. However this assumption fails unless the sequence of deletion requests is chosen independently of the information that the model provider has made public. This is a very strong assumption, because users may wish to delete their data *exactly because of what deployed models reveal about them*.

We give a generic reduction. We show that if:

1. A data deletion algorithm $\mathcal{R}_{\mathcal{A}}$ for a learning algorithm $\mathcal{A}$ has deletion guarantees for *oblivious* sequences of deletion requests (as those from past work do), and

2. Information about the internal randomness of $\mathcal{R}_{\mathcal{A}}$ is revealed only in a manner that satisfies *differential privacy*, then

$(\mathcal{A}, \mathcal{R}_{\mathcal{A}})$ *also satisfies data deletion guarantees against an adaptive sequence of deletion requests*, that can depend in arbitrary ways on the information that the model provider has made public.

In Section 3, we motivate our main result with a theoretical example which illustrates that past method's lack of guarantees for adaptive sequences is not simply a failure of analysis, but an actual failure of these methods to satisfy deletion guarantees for adaptive deletion sequences. As an exemplar, we use a variant of SISA from Bourtoule et al. [2021] that satisfies perfect deletion guarantees for non-adaptive deletion sequences and exhibit adaptive deletion sequences that strongly separate the resulting distribution on models compared to the retraining baseline.

The generic reduction found in Section 4 can be used to give adaptive data deletion mechanisms for a wide variety of problems by leveraging past work on deletion algorithms for non-adaptive sequences, and a line of work on differentially private aggregation [Papernot et al., 2018, Dwork and Feldman, 2018]. Since prior deletion algorithms themselves tend to use existing learning algorithms in a black-box way, the entire pipeline is modular and easy to bolt-on to existing methods. In Section 5, we show how this can be accomplished by using a variant of the SISA framework of Bourtoule et al. [2021] together with a differentially private aggregation method.

In Section 6, we complement our main result with a set of experimental results on CIFAR-10, MNIST, and Fashion-MNIST that demonstrate differential privacy may be useful in giving adaptive guarantees beyond the statement of our theorems. Specifically we show that small amounts of noise addition (insufficient for our theorems to apply) already serve to break the adaptive deletion strategies that we use to falsify the adaptive deletion guarantees in our experiments described in Section 3 and do so at minimal expense in model accuracy.

## 1.2 Related Work

Data deletion was introduced by Cao and Yang [2015]; we adopt the randomized formulation of Ginart et al. [2019]. Ginart et al. [2019] anticipate the problem of deletion requests that might be correlated with internal state of the algorithm, and define (and propose as a study for future work) *robust* data deletion which is a data deletion guarantee that holds for adversaries with knowledge of the internal state. Our insight is that we can provide deletion guarantees against adaptive sequences by instead obscuring the internal state of the algorithm using techniques from differential privacy.

We are the first to explicitly consider the problem of adaptive sequences of deletion requests, but some techniques from past work *do* have deletion guarantees that extend to adaptive sequences. Deterministic methods and methods that depend only on randomness that is sampled after the deletion request are already robust to adaptive deletion. This includes techniques that find an approximately optimal solution to a strongly convex problem and then perturb the solution to obscure the optimizer within a small radius e.g. Guo et al. [2019], Neel et al. [2021], Sekhari et al. [2021]. It also includes the approach of Golatkar et al. [2020a,b] which pre-trains a nonconvex model on data that will never be deleted and then does convex fine-tuning on user data on top of that. Techniques whose deletion guarantees depend on randomness sampled at training in general do not have guarantees against adaptive deletions. This includes algorithms given in Ginart et al. [2019], Bourtoule et al. [2021], Neel et al. [2021] — the SISA framework of Bourtoule et al. [2021] being of particular interest as it is agnostic to the class of models and training methodology, and so is extremely flexible.

Differential privacy has been used as a mitigation for adaptivity since the work of Dwork et al. [2015c,a]. In machine learning, it has been used to mitigate the bias of adaptive data gathering strategies as used in bandit learning algorithms [Neel and Roth, 2018]. The application that is most similar to our work is Hassidim et al. [2020], which uses differential privacy of the internal randomness of an algorithm (as we do) to reduce streaming algorithms with guarantees against adaptive adversarial streams to streaming algorithms with guarantees against oblivious adversaries. Our techniques differ; while Hassidim et al. [2020] reduce to the so-called "transfer theorem for linear and low sensitivity queries" developed over a series of works Dwork et al. [2015c], Bassily et al. [2021], Jung et al. [2020], we use a more general connection between differential privacy and "max-information" established in Dwork et al. [2015b], Rogers et al. [2016].

## 2 Preliminaries

Let $\mathcal{Z}$ be the data domain. A dataset $D$ is a multi-set of elements from $\mathcal{Z}$. We consider update requests of two types: deletion and addition. These update requests are formally defined below, similar to how they are defined in [Neel et al., 2021].

**Definition 2.1** (Update Operations and Sequences). *An update $u$ is a pair $(z, \bullet)$ where $z \in \mathcal{Z}$ is a datapoint and $\bullet \in \mathcal{T} = \{'\texttt{add}', '\texttt{delete}'\}$ determines the type of the update. An update sequence $U$ is a sequence $(u^1, u^2, \ldots)$ where $u^t \in \mathcal{Z} \times \mathcal{T}$ for all $t$. Given a dataset $D$ and an update $u = (z, \bullet)$, the update operation is defined as:*

$$D \circ u \triangleq \begin{cases} D \cup \{z\} & \textit{if } \bullet = '\texttt{add}' \\ D \setminus \{z\} & \textit{if } \bullet = '\texttt{delete}' \end{cases}$$

*Given an update sequence $U = (u^1, u^2, \ldots)$, we have $D \circ U \triangleq (((D \circ u^1) \circ u^2) \circ \ldots)$.*

We use $\Theta$ to denote the space of models. A *learning* or *training* algorithm is a mapping $\mathcal{A} : \mathcal{Z}^* \to \Theta^*$ that maps a dataset $D \in \mathcal{Z}^*$ to a collection of models $\theta \in \Theta^*$. An *unlearning* or *update* algorithm for $\mathcal{A}$ is a mapping $\mathcal{R}_{\mathcal{A}} : \mathcal{Z}^* \times (\mathcal{Z} \times \mathcal{T}) \times \mathcal{S} \to \Theta^*$ which takes in a data set $D \in \mathcal{Z}^*$, an update request $u \in \mathcal{Z} \times \mathcal{T}$, and some current state for the algorithms $s \in \mathcal{S}$ (the domain $\mathcal{S}$ can be arbitrary), and outputs an updated collection of models $\theta' \in \Theta^*$. In this paper we consider a setting in which a stream of update requests arrive in sequence. We note that in this sequential framework, the update algorithm $\mathcal{R}_{\mathcal{A}}$ also updates the state of the algorithm after each update request is processed; however, for notational economy, we do not explicitly write the updated state as an output of the algorithm.

At each round, we provide access to the models through a mapping $f_{\text{publish}}^t : \Theta^* \to \Psi$ that takes in the collection of models and outputs some object $\psi \in \Psi$. A published object $\psi \in \Psi$ can, for instance, be the aggregate predictions of the learned models on a data set, or, some aggregation of the models. To model adaptively chosen update sequences, we define an arbitrary "update requester" who interacts with the learning and unlearning algorithms $(\mathcal{A}, \mathcal{R}_{\mathcal{A}})$ through the publishing function $f_{\text{publish}}$ in rounds to generate a sequence of updates. The update requester is denoted by `UpdReq` and defined in Definition 2.2, and the interaction between the algorithms and the update requester is described in Algorithm 1.

Throughout we will use $u^t$ to denote the update request at round $t$. We will use $D^t$ to denote the data set at round $t$: $D^0$ is the initial training data set and for all $t \geq 1$, $D^t = D^{t-1} \circ u^t$. We will use $\theta^t$ to denote the learned models at round $t$: $\theta^0$ is generated by the initial training algorithm $\mathcal{A}$, and $\theta^t$ for

**Algorithm 1:** Interaction between $(\mathcal{A}, \mathcal{R}_{\mathcal{A}})$ and `UpdReq`

---

1: **Input**: Data set $D$
2: Let $D^0 \leftarrow D$.
3: Train $\theta^0 \leftarrow \mathcal{A}(D)$.
4: Publish $\psi^0 \leftarrow f_{\text{publish}}^0(\theta^0)$.
5: Save the initial state $s^0$.
6: **for** $t = 1, 2, \ldots$ **do**
7:     The update requester requests a new update, given the history of interaction:
8:         $u^t \leftarrow \texttt{UpdReq}\left(\psi^0, u^1, \psi^1, u^2, \ldots, u^{t-1}, \psi^{t-1}\right)$.
9:     The algorithms update, given $u^t$:
10:         Update the models $\theta^t \leftarrow \mathcal{R}_{\mathcal{A}}\left(D^{t-1}, u^t, s^{t-1}\right)$.
11:         Publish $\psi^t \leftarrow f_{\text{publish}}^t(\theta^t)$.
12:         Save the updated state $s^t$.
13:         Update the data set $D^t \leftarrow D^{t-1} \circ u^t$.

---

$t \geq 1$ denotes the updated models at round $t$ generated by the update algorithm $\mathcal{R}_{\mathcal{A}}$. $\psi^t$ denotes the published object at round $t$: $\psi^t = f_{\text{publish}}^t(\theta^t)$.

**Definition 2.2** (Update Requester (`UpdReq`)). *The update sequence is generated by an update requester which is modeled by a (possibly randomized) mapping* $\texttt{UpdReq} : \Psi^* \times (\mathcal{Z} \times \mathcal{T})^* \rightarrow (\mathcal{Z} \times \mathcal{T})$ *that takes as input the history of interaction between herself and the algorithms, and outputs a new update for the current round. Given an update requester* `UpdReq`, *algorithms* $(\mathcal{A}, \mathcal{R}_{\mathcal{A}})$ *and publishing functions* $\{f_{publish}^t\}_t$, *the update sequence* $U = \{u^t\}_t$ *can be written as*

$$u^1 = \texttt{UpdReq}\left(\psi^0\right), \ u^2 = \texttt{UpdReq}\left(\psi^0, u^1, \psi^1\right), \ldots, \ u^t = \texttt{UpdReq}\left(\psi^0, u^1, \psi^1, \ldots, u^{t-1}, \psi^{t-1}\right)$$

*We say an update requester* `UpdReq` *is nonadaptive if it is independent of the published objects, i.e., if there exists a mapping* $\texttt{UpdReq}' : (\mathcal{Z} \times \mathcal{T})^* \rightarrow (\mathcal{Z} \times \mathcal{T})$ *such that for all* $t \geq 1$,

$$u^t = \texttt{UpdReq}\left(\psi^0, u^1, \psi^1, u^2, \ldots, u^{t-1}, \psi^{t-1}\right) = \texttt{UpdReq}'\left(u^1, u^2, \ldots, u^{t-1}\right)$$

*This is equivalent to saying that the update sequence is fixed before the interaction occurs.*

Following [Ginart et al., 2019], we propose the following definition for an unlearning algorithm in the sequential update setting ([Ginart et al., 2019] gives a definition for a single deletion request, whereas here we define a natural extension for an arbitrarily long sequence of deletions, as well as additions, that can be chosen adaptively.). Informally, we require that at every round, and for all possible update requesters, with high probability over the draw of the update sequence, no subset of models resulting from deletion occurs with substantially higher probability than it would have under full retraining.

**Definition 2.3** ($(\alpha, \beta, \gamma)$-unlearning). *We say that* $\mathcal{R}_{\mathcal{A}}$ *is an* $(\alpha, \beta, \gamma)$-*unlearning algorithm for* $\mathcal{A}$, *if for all datasets* $D = D^0$ *and all update requesters* `UpdReq`, *the following condition holds: For every update step* $t \geq 1$, *with probability at least* $1 - \gamma$ *over the draw of the update sequence* $u^{\leq t} = (u^1, \ldots, u^t)$ *from* `UpdReq`,

$$\forall E \subseteq \Theta^* : \quad \Pr\left[\mathcal{R}_{\mathcal{A}}\left(D^{t-1}, u^t, s^{t-1}\right) \in E \,\middle|\, u^{\leq t}\right] \leq e^\alpha \cdot \Pr\left[\mathcal{A}\left(D^t\right) \in E\right] + \beta$$

*We say* $\mathcal{R}_{\mathcal{A}}$ *is a nonadaptive* $(\alpha, \beta, \gamma)$-*unlearning algorithm for* $\mathcal{A}$ *if the above condition holds for any nonadaptive* `UpdReq`.

**Remark 2.1.** *Our definition of unlearning is reminiscent of differential privacy, but following [Ginart et al., 2019], we ask only for a* one-sided *guarantee: that the probability of any event under the unlearning scheme is not too much larger than the probability of the same event under full retraining, but not vice versa. The reason is that we do not want there to be events that can substantially increase an observer's confidence that we did* not *engage in full retraining, but we do not object to observers who strongly update their beliefs that we did* engage in full retraining. *Our events* $E$ *are defined directly over the sets of models in* $\Theta^*$ *output by* $\mathcal{A}$ *and* $\mathcal{R}_{\mathcal{A}}$ — *note that because of information processing inequalities, this is only stronger than defining events* $E$ *over the observable outcome space* $\Psi$.

## 2.1 Differential Privacy and Max-Information

Differential privacy will be a key tool in our results. Let $\mathcal{X}$ denote an arbitrary data domain. We use $x \in \mathcal{X}$ to denote an individual element of $\mathcal{X}$, and $X \in \mathcal{X}^*$ to denote a collection of elements from $\mathcal{X}$ — which we call a data set. We say two data sets $X, X' \in \mathcal{X}^*$ are neighboring if they differ in at most one element. We say an algorithm $M : \mathcal{X}^n \to \mathcal{O}$ is differentially private if its output distributions on neighboring data sets are close, formalized below.

**Definition 2.4** (Differential Privacy (DP) [Dwork et al., 2006b,a]). *An algorithm $M : \mathcal{X}^m \to \mathcal{O}$ is $(\epsilon, \delta)$-differentially private, if for every neighboring $X$ and $X'$, and for every $O \subseteq \mathcal{O}$, we have* $\Pr[M(X) \in O] \leq e^\epsilon \Pr[M(X') \in O] + \delta$.

We remark at the outset that the "datasets" to which we will eventually ask for differential privacy with respect to will not be the datasets on which our learning algorithms are trained, but will instead be collections of random bits parameterizing our randomized algorithms.

Differentially private algorithms are robust to data-independent post-processing:

**Lemma 2.1** (Post-processing preserves DP [Dwork et al., 2006b]). *If $M : \mathcal{X}^m \to \mathcal{O}$ is $(\epsilon, \delta)$-differentially private, then for all $f : \mathcal{O} \to \mathcal{R}$, we have $f \circ M : \mathcal{X}^m \to \mathcal{R}$ defined by $f \circ M(X) = f(M(X))$ is $(\epsilon, \delta)$-differentially private.*

The max-information between two jointly distributed random variables measures how close their joint distribution is to the product of their corresponding marginal distributions.

**Definition 2.5** (Max-Information [Dwork et al., 2015b]). *Let $X$ and $Y$ be jointly distributed random variables over the domain $(\mathcal{X}, \mathcal{Y})$. The $\beta$-approximate max-information between $X$ and $Y$ is:*

$$I_\infty^\beta(X; Y) = \log \sup_{E \subseteq (\mathcal{X}, \mathcal{Y}), \Pr[(X,Y) \in E] > \beta} \frac{\Pr[(X, Y) \in E] - \beta}{\Pr[(X \otimes Y) \in E]}$$

*where $(X \otimes Y)$ represents the product distribution of $X$ and $Y$.*

The max-information of an algorithm $M$ that takes a dataset $X$ as input and outputs $M(X)$, is defined as the max-information between $X$ and $M(X)$ for the worst case product distribution over $X$:

**Definition 2.6** (Max-Information of an Algorithm [Dwork et al., 2015b]). *Let $M : \mathcal{X}^m \to \mathcal{O}$ be an Algorithm. We say $M$ has $\beta$-approximate max-information of $k$, written $I_\infty^\beta(M, m) \leq k$, if for every distribution $\mathcal{P}$ over $\mathcal{X}$, we have $I_\infty^\beta(X; M(X)) \leq k$ when $X \sim \mathcal{P}^m$.*

In this paper, we will use the fact that differentially private algorithms have bounded max-information:

**Theorem 2.1** (DP implies bounded max-information [Rogers et al., 2016]). *Let $M : \mathcal{X}^m \to \mathcal{O}$ be an $(\epsilon, \delta)$-differentially private algorithm for $0 < \epsilon \leq 1/2$ and $0 < \delta < \epsilon$. Then, $I_\infty^\beta(M, m) = O\left(\epsilon^2 m + m\sqrt{\delta/\epsilon}\right)$ for $\beta = e^{-\epsilon^2 m} + O\left(m\sqrt{\delta/\epsilon}\right)$.*

## 3 Falsifying Unlearning Guarantees with Adaptivity

In this section we demonstrate that the deletion guarantees of algorithms in the SISA framework [Bourtoule et al., 2021] fail for adaptive deletion sequences. We give a clean toy construction which shows algorithms in the SISA framework fail to have nontrivial adaptive deletion guarantees even in the black-box setting when the models within each shard are not made public, only aggregations of their classification outputs. In the Appendix we experimentally evaluate a more realistic instantiation of this construction.

The setting we consider directly corresponds to the setting in which our final algorithms operate: what is made public is the aggregate predictions of the ensemble of models, but not the models themselves. For non-adaptive sequences of deletions, distributed algorithms of the sort described in Section 5 have perfect deletion guarantees. We demonstrate via a simple example that these guarantees dramatically fail for adaptive deletion sequences.

Suppose we have a dataset consisting of real-valued points with binary labels $\{(x_i, y_i)\}_{i=1}^{2n}$, $x_i \in \mathbb{R}^d$, $y_i \in \{0, 1\}$ in which there are exactly two copies of each distinct training example. Consider a

simplistic classification model, resembling a lookup table, which given a point $x_i$ predicts the label $y_i$ if the model has been trained on $(x_i, y_i)$ and a dummy prediction value "$\perp$" otherwise:

$$f_{\mathcal{D}}(x_i) = \begin{cases} y_i & \text{if } (x_i, y_i) \in \mathcal{D}, \\ \perp & \text{otherwise} \end{cases}$$

Consider what happens when the training algorithm randomly partitions this dataset into three pieces and trains such a model on each partition. This constructs an ensemble which, at query time, predicts the class with the majority vote. On this dataset, the ensemble will predict the labels of roughly $2/3$ of the training points correctly—that is, exactly those points for which the duplicates have fallen into distinct partitions, so that the ensemble gets the majority vote right.

We construct an adaptive adversary who chooses to delete exactly those training points that the ensemble correctly classifies (which are those points for whom the duplicates have fallen into distinct shards). The result is that the model resulting from this deletion sequence will misclassify every remaining training point. Full retraining (because it would rerandomize the partition) would again lead to training accuracy of approximately $2/3$. Recalling that our deletion notion requires that the probability of any event under the unlearning scheme is not much larger than the probability of the same event under full retraining, this demonstrates that there are algorithms in the SISA framework — even if the models are not directly exposed — that do not satisfy $(\alpha, \beta, \gamma)$-deletion guarantees for any nontrivial value of $\alpha$. We formalize this below:

**Theorem 3.1.** *There are learning and unlearning algorithms in the SISA framework $(\mathcal{A}, \mathcal{R}_{\mathcal{A}})$ such that for any $\alpha$, and any $\beta, \gamma < 1/4$, $\mathcal{R}_{\mathcal{A}}$ is not an $(\alpha, \beta, \gamma)$-unlearning algorithm for $\mathcal{A}$.*

A proof of this theorem can be found in the appendix.

# 4  A Reduction from Adaptive to Nonadaptive Update Requesters

In our analysis we imagine without loss of generality that the learning algorithm $\mathcal{A}$ draws an *i.i.d.* sequence of random variables $r \sim \mathcal{P}^m$ (that encodes all the randomness to be used over the course of the updates) from some distribution $\mathcal{P}$, and passes it to the unlearning algorithm $\mathcal{R}_{\mathcal{A}}$. Note $r$ is drawn once in the initial training, and given $r$, $\mathcal{A}$ and $\mathcal{R}_{\mathcal{A}}$ become deterministic mappings. We can also view the state $s^t$ as a deterministic mapping of $r$, the update requests so far $u^{\leq t} = (u^1, \ldots, u^t)$, and the original data set $D^0$. We write $s^t = g^t(D^0, u^{\leq t}, r)$ for some deterministic mapping $g^t$. We can therefore summarize the trajectory of the algorithms $(\mathcal{A}, \mathcal{R}_{\mathcal{A}})$ as follows.

- $t = 0$: draw $r \sim \mathcal{P}^m$, let $\theta^0 = \mathcal{A}(D) \equiv \mathcal{A}(D; r)$, and $\psi^0 = f_{\text{publish}}^0\left(\theta^0\right)$.
- $t \geq 1$: $\theta^t = \mathcal{R}_{\mathcal{A}}(D^{t-1}, u^t, s^{t-1})$ where $s^{t-1} = g^{t-1}(D^0, u^{\leq t-1}, r)$, and $\psi^t = f_{\text{publish}}^t\left(\theta^t\right)$.

In this view, the randomness $r$ used by the learning algorithm $\mathcal{A}$ and the subsequent invocations of the unlearning algorithm $\mathcal{R}_{\mathcal{A}}$ is represented as part of the internal state. Past analyses of unlearning algorithms have crucially assumed that $r$ is statistically independent of the updates $(u^1, u^2, \ldots)$ (which is the case for non-adaptive update requesters, but not for adaptive update requesters). In the following general theorem, we show that if a learning/unlearning pair satisfies unlearning guarantees against non-adaptive update requesters, and the publishing function is differentially private *in the internal randomness $r$*, then the resulting algorithms also satisfy unlearning guarantees against adaptive update requesters. Note that what is important is that the publishing algorithms are differentially private in the *internal randomness $r$*, not in the datapoints used for training.

**Theorem 4.1** (A General Theorem). *Fix a pair of learning and unlearning algorithms $(\mathcal{A}, \mathcal{R}_{\mathcal{A}})$ and the publishing functions $\{f_{\text{publish}}^t\}_t$. Suppose for every round $t$, the sequence of publishing functions $\{f_{\text{publish}}^{t'}\}_{t' \leq t}$ is $(\epsilon, \delta)$-differentially private in $r \sim \mathcal{P}^m$, for $0 < \epsilon \leq 1/2$ and $0 < \delta < \epsilon$. Suppose $\mathcal{R}_{\mathcal{A}}$ is a non-adaptive $(\alpha, \beta, \gamma)$-unlearning algorithm for $\mathcal{A}$. Then $\mathcal{R}_{\mathcal{A}}$ is an $(\alpha', \beta', \gamma')$-unlearning algorithm for $\mathcal{A}$ for $\alpha' = \alpha + \epsilon'$, $\beta' = \beta e^{\epsilon'} + \sqrt{\delta'}, \gamma' = \gamma + \sqrt{\delta'}$ where $\epsilon' = O\left(\epsilon^2 m + m\sqrt{\delta/\epsilon}\right)$ and $\delta' = e^{-\epsilon^2 m} + O\left(m\sqrt{\delta/\epsilon}\right)$.*

The proof can be found in the Appendix, but at an intuitive level, it proceeds as follows. Because it does not change the joint distribution on update requests and internal state, we can imagine in our

---

**Algorithm 2:** $\mathcal{A}^{\text{distr}}$: Distributed Learning Algorithm

---

**Input**: dataset $D \equiv D^0$ of size $n$
Draw the shards: $D_i^0 = \texttt{Sampler}(D^0, p)$, for every $i \in [k]$.
Train the models: $\theta_i^0 = \mathcal{A}^{\text{single}}(D_i^0)$, for every $i \in [k]$.
Save the state: $s^0 = (\{D_i^0\}_{i \in [k]}, \{\theta_i^0\}_{i \in [k]})$ `// to be used for the 1st update.`
**Output**: $\{\theta_i^0\}_{i \in [k]}$

---

analysis that $r$ is redrawn after each update request from its conditional distribution, conditioned on the observed update sequence so far. Because the publishing function is differentially private in $r$, by the fact that post-processing preserves differential privacy (Lemma 2.1), so is the update sequence. We may therefore apply the max-information bound (Theorem 2.1), which allows us to relate the conditional distribution on $r$ to its original (prior) distribution $\mathcal{P}^m$. But resampling $r$ from $\mathcal{P}^m$ removes the dependence between $r$ and the update sequence, which places us in the non-adaptive case, and allows us to apply the hypothesized unlearning guarantees for nonadaptive update requesters.

## 5 Distributed Algorithms

In this section, we describe a general family of distributed learning and unlearning algorithms that are in the spirit of the "SISA" framework of Bourtoule et al. [2021] (with one crucial modification). At a high level, the SISA framework operates by first randomly dividing the data into $k$ "shards", and separately training a model on each shard. When a new point is deleted, it is removed from the shards that contained it, and only the models corresponding to those shards are retrained. The flexibility of this methodology is that the models and training procedures used in each shard can be arbitrary, as can the aggregation done at the end to convert the resulting ensemble into predictions: however these choices are instantiated, this framework gives a $(0, 0, 0)$-unlearning algorithm against any non-adaptive update requester (Lemma 5.1). Here we show that if the $k$ shards are selected *independently* of one another, then we can apply our reduction given in the previous section with $m = k$ and obtain algorithms that satisfy deletion guarantees against adaptive update requesters.

A *distributed* learning algorithm $\mathcal{A}^{\text{distr}} : \mathcal{Z}^* \to \Theta^*$ is described by a single-shard learning algorithm $\mathcal{A}^{\text{single}} : \mathcal{Z}^* \to \Theta$ and a routine $\texttt{Sampler}$, used to select the points in a shard. $\texttt{Sampler}$, given a dataset $D$ and some probability $p \in [0, 1]$, includes each element of $D$ in the shard with probability $p$.

Distributed learning algorithm $\mathcal{A}^{\text{distr}}$ creates $k$ independent shards from the dataset $D$ of size $n$ by running $\texttt{Sampler}$ $k$ times and training a model with $\mathcal{A}^{\text{single}}$ on each shard $i \in [k]$ to form an ensemble of $k$ models. To emphasize that the randomness across shards is independent, we will instantiate $k$ independent samplers $\texttt{Sampler}_i$ and training algorithms $\mathcal{A}_i^{\text{single}}$ for each shard $i \in [k]$. We formally describe $\mathcal{A}^{\text{distr}}$ in Algorithm 2.

The state $s$ of the unlearning algorithm $\mathcal{R}_{\mathcal{A}^{\text{distr}}}$ records the $k$ shards $\{D_i\}_i$ and the ensemble of $k$ models $\{\theta_i\}_i$. Thus $\mathcal{S} = \{\mathcal{Z}^*\}^k \times \Theta^k$. As an update request $u$ is received, the update function removes the data point from every shard that contains it (for deletion) or adds the new point to each shard with probability $p$ (for addition). In either case, only the models corresponding to shards that have been updated are retrained using $\mathcal{A}^{\text{single}}$. We formally describe $\mathcal{R}_{\mathcal{A}^{\text{distr}}}$ in Algorithm 3.

First, we show that if the update requester is non-adaptive, $\mathcal{R}_{\mathcal{A}^{\text{distr}}}$ is a $(0, 0, 0)$-unlearning algorithm:

**Lemma 5.1.** $\mathcal{R}_{\mathcal{A}^{\text{distr}}}$ *is a non-adaptive* $(0, 0, 0)$-*unlearning algorithm for* $\mathcal{A}^{\text{distr}}$.

Now, by combining Lemma 5.1 and our general Theorem 4.1, we can show the following:

**Theorem 5.1** (Unlearning Guarantees)**.** *If for every round $t$, the sequence of publishing functions $\{f_{\text{publish}}^{t'}\}_{t' \leq t}$ is $(\epsilon, \delta)$-differentially private in the random seeds $r \sim \mathcal{P}^k$ of the algorithms for $0 < \epsilon \leq 1/2$ and $0 < \delta < \epsilon$, then $\mathcal{R}_{\mathcal{A}^{\text{distr}}}$ is an $(\alpha, \beta, \gamma)$-unlearning algorithm for $\mathcal{A}^{\text{distr}}$ where*

$$\alpha = O\left(\epsilon^2 k + k\sqrt{\delta/\epsilon}\right), \quad \beta = \gamma = O\left(\sqrt{e^{-\epsilon^2 k} + k\sqrt{\delta/\epsilon}}\right)$$

Next, we bound the time complexity of our algorithms:

**Algorithm 3:** $\mathcal{R}_{\mathcal{A}^{\text{distr}}}$: Distributed Unlearning Algorithm: $t$'th round of unlearning

---

**Input**: dataset $D^{t-1}$, update $u^t = (z^t, \bullet^t)$, state $s^{t-1} = (\{D_i^{t-1}\}_{i \in [k]}, \{\theta_i^{t-1}\}_{i \in [k]})$

**if** $\bullet^t = \text{'delete'}$ **then**
  $S = \{i \in [k] : z^t \in D_i^{t-1}\}$ // the shards $z^t$ belongs to.
**else**
  $S = \{i \in [k] : \texttt{Sampler}_i(\{z^t\}, p) \neq \{\}\}$ // the shards $z^t$ will be added to.

Update the shards: $D_i^t = \begin{cases} D_i^{t-1} \circ u^t & \text{if } i \in S \\ D_i^{t-1} & \text{otherwise} \end{cases}$, for every $i \in [k]$.

Update the models: $\theta_i^t = \begin{cases} \mathcal{A}^{\text{single}}(D_i^t) & \text{if } i \in S \\ \theta_i^{t-1} & \text{otherwise} \end{cases}$, for every $i \in [k]$.

Update the state: $s^t = (\{D_i^t\}_{i \in [k]}, \{\theta_i^t\}_{i \in [k]})$ // to be used for the next update.

**Output**: $\{\theta_i^t\}_{i \in [k]}$

---

**Theorem 5.2** (Run-time Guarantees). *Let $p = 1/k$. Suppose the publishing functions satisfy the differential privacy requirement of Theorem 5.1. Let $N^t$ denote the number of times $\mathcal{R}_{\mathcal{A}}^{distr}$ calls $\mathcal{A}^{single}$ at round $t$. We have that $N^0 = k$, and for every round $t \geq 1$: 1) if the update requester is non-adaptive, for every $\xi$, with probability at least $1 - \xi$, $N^t \leq 1 + \sqrt{2\log(1/\xi)}$. 2) if the update requester is adaptive, for every $\xi$, with probability at least $1 - \xi$, $N^t \leq 1 + \sqrt{2\log((n+t)/\xi)}$. Furthermore, for $\xi > \delta'$, with probability at least $1 - \xi$, we have*

$$N^t \leq 1 + \min\left\{\sqrt{2\log(2(n+t)/(\xi - \delta'))}, \sqrt{2\epsilon' + 2\log(2/(\xi - \delta'))}\right\}$$

*where $\epsilon' = O\left(\epsilon^2 k + k\sqrt{\delta/\epsilon}\right)$ and $\delta' = e^{-\epsilon^2 k} + O\left(k\sqrt{\delta/\epsilon}\right)$*

The proof can be found in the appendix, but at a high level it proceeds as follows. For a deletion request, we must retrain every shard that contains the point to be deleted. For a non-adaptive deletion request, we retrain one shard in expectation and we can obtain a high probability upper bound by using a Hoeffding bound. In the adaptive case, this may no longer be true, but there are two ways to obtain upper bounds that correspond to the two bounds in our Theorem. We can provide a worst-case upper bound on the number of shards that *any* of the $n$ data points belongs to, which incurs a cost of order $\sqrt{\log n}$. Alternately, we can apply max-information bounds to reduce to the non-adaptive case, using an argument that is similar to our reduction for deletion guarantees.

**Remark 5.1.** *We note that there is an alternative algorithm that one might consider, resulting from group differential privacy. If a learning algorithm satisfies $\frac{\epsilon}{k}$−differential privacy, a valid unlearning procedure is to do nothing for $k$ updates and then fully retrain on the $(k+1)^{th}$ update. This follows from the $\epsilon$−differential privacy guarantee the algorithm will have for groups of size $k$. Our algorithm substantially outperforms this alternative algorithm as well, namely because our privacy parameter degrades much slower than in this group privacy baseline. Our analysis leverages adaptive composition of privacy across the publishing functions which means that privacy degrades with the square root of the number of updates, while it degrades linearly with group privacy. Consequently the group privacy baseline would require a full retraining every $k$ updates, but our algorithm requires a full retraining only every $k^2$ updates.*

## 5.1 Private Aggregation

We briefly describe how we serve prediction requests by privately aggregating the output of the ensemble of models such that the published predictions are differentially private in the random seeds $r$. At each round $t$, while $\mathcal{R}_{\mathcal{A}}^{\text{distr}}$ is waiting for the next update request $u^{t+1}$, we receive prediction requests $x$ and serve predictions $\hat{y}$. For each prediction request, we privately aggregate the predictions made by the ensemble of models $\{\theta_i^t\}_i$; Dwork and Feldman [2018] show several ways to privately aggregate predictions (one simple technique is to use the exponential mechanism to approximate the majority vote). Suppose we aggregate the predictions made by the ensemble of models using $\texttt{PrivatePredict}_{\epsilon'}^k : \Theta^k \times \mathcal{X} \to \mathcal{Y}$, which takes in an ensemble of $k$ models and a data point, aggregates predictions from the ensemble models, and outputs a label that is $\epsilon'$-differentially private

in the models. If we receive $l^t$ many prediction requests $(x_1^t, \ldots, x_{l^t}^t)$ before our next update request $u^{t+1}$, we can write $(\hat{y}_1^t, \ldots, \hat{y}_{l^t}^t) = f_{\text{publish}}^t(\{\theta_i^t\}_i)$ where $\hat{y}_j^t = \texttt{PrivatePredict}_{\epsilon'}^k(\{\theta_i^t\}_i, x_j^t)$.

Theorem 5.1, tells us that desired unlearning parameters $(\alpha, \beta, \gamma)$ can be obtained by guaranteeing that the sequence of predictions is $(\epsilon, \delta)$ differentially private in the models (and hence $r$), for target parameters $\epsilon, \delta$. As we serve prediction requests using $\texttt{PrivatePredict}_{\epsilon'}^k$, our privacy loss will accumulate and eventually exhaust our budget of $(\epsilon, \delta)$-differential privacy. Hence we must track our accumulated privacy loss in the state of our unlearning algorithm, and when it is exhausted, fully retrain using $\mathcal{A}^{\text{distr}}$. This resamples $r$ and hence resets our privacy budget. Standard composition theorems (see Dwork and Roth [2014]) show that we exhaust our privacy budget (and need to fully retrain) every time the number of prediction requests made since the last full retraining exceeds $\left\lfloor \frac{\epsilon^2}{8(\epsilon')^2 \ln(\frac{1}{\delta})} \right\rfloor$.

We formally describe this process denoted as $\texttt{PrivatePredictionInteraction}(\epsilon', \epsilon, \delta, k)$ in the appendix and state its unlearning guarantee in Theorem 5.3.

**Theorem 5.3.** *The models* $\{\{\theta_i^t\}_i\}_t$ *in* $\texttt{PrivatePredictionInteraction}(\epsilon', \epsilon, \delta, k)$ *satisfy* $(\alpha, \beta, \gamma)$-*unlearning guarantee for* $\mathcal{A}^{\text{distr}}$ *where* $\alpha = O\left(\epsilon^2 k + k\sqrt{\delta/\epsilon}\right)$ *and* $\beta, \gamma = O\left(\sqrt{e^{-\epsilon^2 k} + k\sqrt{\delta/\epsilon}}\right)$, *if* $0 < \epsilon \leq 1/2$ *and* $0 < \delta < \epsilon$.

# 6 Evaluation of Unlearning Guarantees

In this section we consider the white-box setting in which the models in each shard are made public. SISA continues to have perfect deletion guarantees against *non-adaptive* deletion sequences in this setting. Experimental results on CIFAR-10 [Krizhevsky and Hinton, 2009], MNIST [Lecun et al., 1998], and Fashion-MNIST [Xiao et al., 2017] show both the failure of SISA to satisfy adaptive deletion guarantees, and give evidence that differential privacy can mitigate this problem well beyond the setting of our theorems while achieving accuracy only modestly worse than SISA. The code for our experiments can be found at `https://github.com/ChrisWaites/adaptive-machine-unlearning`.

We train SISA with an ensemble of convolutional neural networks on several datasets of points with categorical labels. Given a new point at query time, each model in the ensemble votes on the most likely label and aggregates their votes. The models are exposed publicly. This scheme has perfect non-adaptive deletion guarantees.

To construct an adaptive deletion sequence to falsify the hypothesis that the scheme has adaptive deletion guarantees, we exploit the observation that neural networks are often *overconfident* in the correct label for points on which they have been trained. For each training point, we guess that it falls into the shard corresponding to the model that has the highest confidence for the correct label. We then delete points for which we guess that they fall into the first $k/2$ of the shards, and do not delete any others. After deleting the targeted points, we compute a test statistic: the indicator of whether the average accuracy of the models from the targeted shards is lower than the average accuracy of the models from the non-targeted shards. Under full retraining, by the symmetry of the random partition, the expectation of this test statistic is 0.5. Thus under the null hypothesis that the deletion algorithm satisfies perfect deletion guarantees, the test statistic also has expectation 0.5. Therefore, to the extent that the expectation of the indicator differs from 0.5, we falsify the null hypothesis that SISA has adaptive data deletion guarantees, and larger deviations from 0.5 falsify weaker deletion guarantees.

We run this experiment on three datasets (CIFAR-10, MNIST, and Fashion-MNIST), and plot the results in Figure 1. We then repeat the experiment by adding various amounts of noise to the gradients in the model training process to guarantee finite levels of differential privacy (though much weaker privacy guarantees than would be needed to invoke our theorems). We observe that on each dataset, modest amounts of noise are sufficient to break our attack (i.e. 95% confidence intervals for the expectation of our indicator include 0.5, and hence fail to falsify the null hypothesis) while still approaching the accuracy of our models trained without differential privacy. This is also plotted in Figure 1. This gives evidence that differential privacy can improve deletion guarantees in the presence of adaptivity even in regimes beyond which our theory gives nontrivial guarantees.

Full experimental details can be found in the appendix.

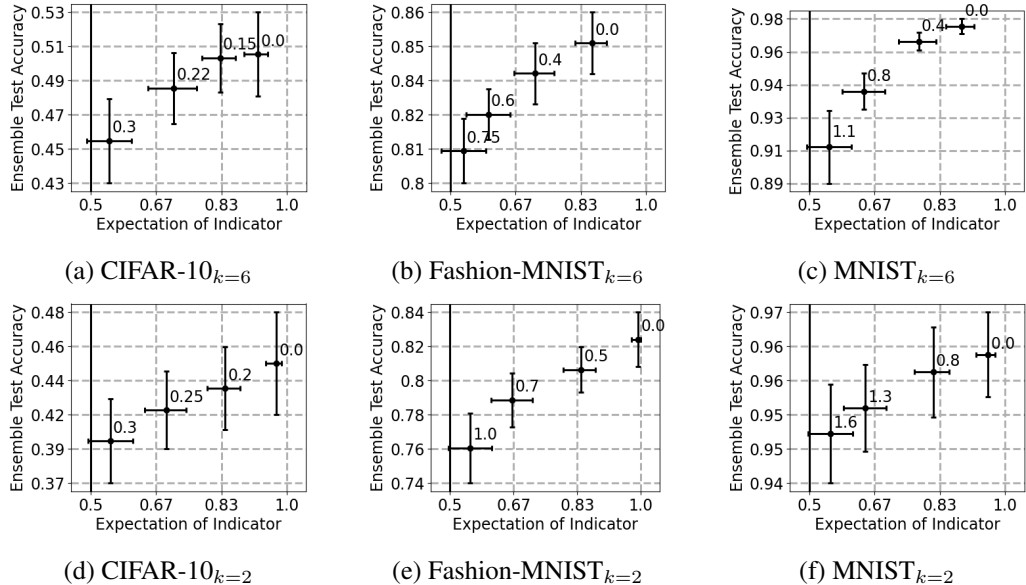

Figure 1: The top row and bottom row show experiments with $k = 6$ and $k = 2$ shards respectively. The 3 columns report on 3 datasets. The $x$ axis denotes estimated expectation of our test statistic (the null hypothesis is expectation $0.5$). The $y$ axis denotes the accuracy of the ensemble after deletion. Each point is annotated with the noise multiplier used in DP-SGD, the standard deviation of Gaussian noise applied to gradients during training. A label of $0.0$ for a point represents the baseline case of no noise (original SISA algorithm). Points are affixed with 95% confidence intervals along both axes (over the randomness of repeating the training/deletion experiment). Horizontal confidence intervals that overlap the line denoting expectation $0.5$ fail to reject the null hypothesis that the algorithm has adaptive data deletion guarantees at $p \leq 0.05$. We get to this point with a level of noise addition that results in only a modest degradation in ensemble performance compared to SISA.

## 7    Conclusion and Discussion

We identify an important blindspot in the data deletion literature (the tenuous implicit assumption that deletion requests are independent of previously released models), and provide a very general methodology to reduce adaptive deletion guarantees to oblivious deletion guarantees. Through this reduction we get the first model and training algorithm agnostic methodology that allows for deletion of arbitrary sequences of adaptively chosen points while giving rigorous guarantees. The constants that our theorems inherit from the max information bounds of Rogers et al. [2016] are such that in most realistic settings they will not give useful parameters. But we hope that these constants will be improved in future work, and we give empirical evidence that differential privacy mitigates adaptive deletion "attacks" at very practical levels, beyond the promises of our theoretical results. We note that like for differential privacy, the $(\alpha, \beta, \gamma)$-deletion guarantees we give in this paper are *parameterized*, and are not meaningful absent a specification of those parameters. There is a risk with such technologies that they will be used with large values of the parameters that give only very weak guarantees, but will be described publicly in a way that glosses over this issue. We therefore recommend that if adopted in deployed products, deletion guarantees always be discussed in public in a way that is precise about what they promise, including the relevant parameter settings.

## Acknowledgements

V.G., C.J., A.R., and S.S. were supported in part by NSF grants CCF-1934876 and AF-1763307, and a grant from the Simons Foundation.

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
