## A  Proof of Theorem 4.1

We first state the following Lemma which we will use to prove Theorem 4.1.

**Lemma A.1** ([Rogers et al., 2016]). *Let $M : \mathcal{X}^m \to \mathcal{O}$ be an $(\epsilon, \delta)$-differentially private algorithm for $0 < \epsilon \leq 1/2$ and $0 < \delta < \epsilon$. Then,*

$$\Pr_{(x,m') \sim (X, M(X))} \left[ \log \left( \frac{\Pr[X = x, M(X) = m']}{\Pr[X = x] \Pr[M(X) = m']} \right) \geq k \right] \leq \beta$$

*where the probability is taken with respect to the joint distribution of $X$ and $M(X)$, and*

$$k = O\left( \epsilon^2 m + m\sqrt{\frac{\delta}{\epsilon}} \right), \quad \beta = e^{-\epsilon^2 m} + O\left( m\sqrt{\frac{\delta}{\epsilon}} \right)$$

**Theorem 4.1** (A General Theorem). *Fix a pair of learning and unlearning algorithms $(\mathcal{A}, \mathcal{R}_\mathcal{A})$ and the publishing functions $\{f_{publish}^t\}_t$. Suppose for every round $t$, the sequence of publishing functions $\{f_{publish}^{t'}\}_{t' \leq t}$ is $(\epsilon, \delta)$-differentially private in $r \sim \mathcal{P}^m$, for $0 < \epsilon \leq 1/2$ and $0 < \delta < \epsilon$. Suppose $\mathcal{R}_\mathcal{A}$ is a non-adaptive $(\alpha, \beta, \gamma)$-unlearning algorithm for $\mathcal{A}$. Then $\mathcal{R}_\mathcal{A}$ is an $(\alpha', \beta', \gamma')$-unlearning algorithm for $\mathcal{A}$ for $\alpha' = \alpha + \epsilon', \beta' = \beta e^{\epsilon'} + \sqrt{\delta'}, \gamma' = \gamma + \sqrt{\delta'}$ where $\epsilon' = O\left( \epsilon^2 m + m\sqrt{\delta/\epsilon} \right)$ and $\delta' = e^{-\epsilon^2 m} + O\left( m\sqrt{\delta/\epsilon} \right)$.*

*Proof.* Fix a data set $D$ and an update requester UpdReq. Fix any unlearning step $t \geq 1$. Note that the sequence of updates up to round $t$, i.e. $u^{\leq t} = (u^1, \ldots, u^t)$, can be seen as a post-processing of the sequence of published objects up to round $t - 1$, i.e. $\psi^{\leq t-1} = (\psi^0, \ldots, \psi^{t-1})$, where the post-processing function is defined by UpdReq (see Definition 2.2). But we know that $\{f_{publish}^{t'}\}_{t' \leq t-1}$ that generates $\psi^{\leq t-1}$ is $(\epsilon, \delta)$-differentially private in $r$. Hence, given that post-processing preserves differential privacy (Lemma 2.1), we have that $u^{\leq t}$ is also $(\epsilon, \delta)$-differentially private in $r$. Consequently, we can apply the fact that DP implies bounded max-information (Lemma A.1) to get that

$$\Pr_{(r,u^{\leq t})} \left[ \log \frac{\Pr[r|u^{\leq t}]}{\Pr[r]} \geq \epsilon' \right] = \Pr_{(r,u^{\leq t})} \left[ \log \frac{\Pr[r, u^{\leq t}]}{\Pr[r]\Pr[u^{\leq t}]} \geq \epsilon' \right] \leq \delta' \qquad (1)$$

where the probability is taken with respect to the joint distribution of $(r, u^{\leq t})$, and that

$$\epsilon' \triangleq O\left( \epsilon^2 m + m\sqrt{\frac{\delta}{\epsilon}} \right), \quad \delta' \triangleq e^{-\epsilon^2 m} + O\left( m\sqrt{\frac{\delta}{\epsilon}} \right)$$

Now define the "Good" event for the update sequence $u^{\leq t}$:

$$G = \left\{ u^{\leq t} : \Pr_{r|u^{\leq t}} \left[ \log \frac{\Pr[r|u^{\leq t}]}{\Pr[r]} \geq \epsilon' \right] \leq \sqrt{\delta'} \right\}$$

We have that

$$\Pr_{u^{\leq t}} \left[ u^{\leq t} \notin G \right] = \Pr_{u^{\leq t}} \left[ \Pr_{r|u^{\leq t}} \left[ \log \frac{\Pr[r|u^{\leq t}]}{\Pr[r]} \geq \epsilon' \right] > \sqrt{\delta'} \right]$$

$$\leq \frac{\mathbb{E}_{u^{\leq t}} \left[ \Pr_{r|u^{\leq t}} \left[ \log \frac{\Pr[r|u^{\leq t}]}{\Pr[r]} \geq \epsilon' \right] \right]}{\sqrt{\delta'}}$$

$$= \frac{\Pr_{(r,u^{\leq t})} \left[ \log \frac{\Pr[r|u^{\leq t}]}{\Pr[r]} \geq \epsilon' \right]}{\sqrt{\delta'}}$$

$$\leq \sqrt{\delta'}$$

where the first inequality is an application of Markov's inequality, and the last one follows from Equation (1). Therefore, if we condition on $\{u^{\le t} \in G\}$ which happens with probability at least $1 - \sqrt{\delta'}$, we have the following guarantee.

$$\Pr_{r|u^{\le t}} \left[ \log \frac{\Pr\left[r|u^{\le t}\right]}{\Pr\left[r\right]} \ge \epsilon' \right] \le \sqrt{\delta'}$$

which in turn implies, with probability $1 - \sqrt{\delta'}$ over the draw of $u^{\le t}$, that for every event $F$ in the space of random seeds $(r)$,

$$\Pr\left[r \in F \mid u^{\le t}\right] \le e^{\epsilon'} \Pr\left[r \in F\right] + \sqrt{\delta'} \tag{2}$$

Now we condition on $\{u^{\le t} \in G\}$. Fix any event $E \subseteq \Theta^*$ in the space of models, and let $F = \{r : \mathcal{R}_{\mathcal{A}}(D^{t-1}, u^t, s^{t-1}) \in E\}$ be the event that the output models of the unlearning algorithm on round $t$ belongs to $E$, recalling that $s^{t-1} = g^{t-1}(D^0, u^{\le t-1}, r)$. Substituting $F$ in Equation (2), we get that

$$\Pr\left[\mathcal{R}_{\mathcal{A}}(D^{t-1}, u^t, s^{t-1}) \in E \mid u^{\le t}\right] \le e^{\epsilon'} \Pr\left[\mathcal{R}_{\mathcal{A}}(D^{t-1}, u^t, s^{t-1}) \in E\right] + \sqrt{\delta'} \tag{3}$$

Note that because on the right hand side we do not condition the probability on the update sequence, we are taking the probability over the distribution of output models of round $t$ for a nonadaptively chosen update sequence. Therefore by the unlearning guarantees for nonadaptive update requesters, we have that with probability at least $1 - \gamma$ over the draw of $u^{\le t}$,

$$\Pr\left[\mathcal{R}_{\mathcal{A}}(D^{t-1}, u^t, s^{t-1}) \in E\right] \le e^{\alpha} \Pr\left[\mathcal{A}(D^t) \in E\right] + \beta \tag{4}$$

Now we can combine Equations (3) and (4) to conclude that, with probability $1 - \gamma - \sqrt{\delta'}$ over $u^{\le t}$,

$$\Pr\left[\mathcal{R}_{\mathcal{A}}(D^{t-1}, u^t, s^{t-1}) \in E \mid u^{\le t}\right] \le e^{\alpha+\epsilon'} \Pr\left[\mathcal{A}(D^t) \in E\right] + \beta e^{\epsilon'} + \sqrt{\delta'}$$

completing the proof. $\qquad \square$

# B  Missing Details from Section 5

**Lemma B.1.** *Consider the distributed learning and unlearning algorithms $\mathcal{A}^{distr}$ and $\mathcal{R}_{\mathcal{A}}^{distr}$. If the update requester is non-adaptive, for every $t$: for every shard $i$, we have $D_i^t$ is an independent draw from the distribution of $\texttt{Sampler}(D^t, p)$.*

*Proof.* We prove this via induction. It's easy to see that this holds true at round $t = 0$ because we explicitly set $D_i^0 = \texttt{Sampler}(D^0, p)$. Now, suppose that $D_i^{\tau-1}$ is an independent draw from the distribution of $\texttt{Sampler}(D^{\tau-1}, p)$ for some $\tau \ge 1$. If the update request $u^\tau = (z^\tau, '\texttt{delete}')$ is a deletion request, then it's easy to see that simply deleting the point $z^\tau$ from every shard that contains it will maintain that each element is chosen to be in the shard with probability $p$. And $D_i^\tau | u^{\tau-1}$ and $D_i^\tau | u^\tau$ must be identically distributed because the update request $u^\tau$ is non-adaptive and has been fixed prior to the interaction — and hence is statistically independent of $D^{\tau-1}$. More formally, we have that for any $z \in D^\tau$,

$$\Pr[z \in D_i^\tau] = \Pr[z \in D_i^\tau | u^{\le \tau}] = \Pr[z \in D_i^\tau | u^{\le \tau-1}] = \Pr[z \in D_i^{\tau-1} | u^{\le \tau-1}] = p.$$

The same argument applies for the addition request where $\mathcal{R}_{\mathcal{A}}^{distr}$ adds the element requested to be added with probability $p$. More formally, we have $\Pr[z \in D_i^\tau] = p$ for any $z \in D^{\tau-1}$ and $\Pr[z^\tau \in D_i^\tau] = p$ by construction. $\qquad \square$

**Lemma 5.1.** *$\mathcal{R}_{\mathcal{A}^{distr}}$ is a non-adaptive $(0, 0, 0)$-unlearning algorithm for $\mathcal{A}^{distr}$.*

*Proof.* Fix any arbitrary round $t \in [T]$. For a non-adaptive $\texttt{UpdReq}$, we can think of the update sequence $u^{\le t}$ as fixed prior to the start of the interaction between the learning procedure and the $\texttt{UpdReq}$. Now, in order to show $(0, 0, 0)$-deletion guarantee of the unlearning algorithm, we need to show that for any $E \subseteq \Theta^*$,

$$\Pr\left[\mathcal{R}_{\mathcal{A}^{distr}}(D^{t-1}, u^t, s^{t-1}) \in E | u^{\le t}\right] = \Pr\left[\mathcal{A}^{distr}(D^t) \in E\right].$$

Note that it is equivalent to show that for any $i \in [k]$ and $E \subseteq \Theta$, we have

$$\Pr\left[\theta_i^t \in E | u^{\leq t}\right] = \Pr\left[\mathcal{A}_i^{\text{single}}(\texttt{Sampler}_i(D^t, p)) \in E\right]$$

because $\texttt{Sampler}_i$ and $\mathcal{A}_i^{\text{single}}$ behave independently across $i \in [k]$ in both $\mathcal{R}_{\mathcal{A}^{\text{distr}}}$ and $\mathcal{A}^{\text{distr}}$. Hence, from here on, we focus on some fixed $i \in [k]$.

Now, we argue that it is sufficient to show that the distribution over $D_i^t$ conditional on $u^{\leq t}$ that is being kept in the state $s^t$ of the unlearning algorithm is exactly the same as that of $\texttt{Sampler}_i(D^t, p)$, which we have already proved in Lemma B.1. Using the fact that update sequence is non-adaptive with respect to the algorithm's randomness, we have for any realization path for shard $i$ until round $t$ (i.e. how the initial shard $D_i^0$ was formed and whether each addition request until round $t$ was actually added to shard $i$ or not)

$$\begin{aligned}
\Pr[\theta_i^t \in E | u^{\leq t}] &= \Pr[\theta_i^{t'} \in E | u^{\leq t}] \\
&= \Pr[\theta_i^{t'} \in E | u^{\leq t'}] \\
&= \Pr[\mathcal{A}_i^{\text{single}}(D_i^{t'}) \in E | u^{\leq t'}] \\
&= \Pr[\mathcal{A}_i^{\text{single}}(D_i^t) \in E | u^{\leq t}]
\end{aligned}$$

where $t' = \min\{\tau \leq t : D_i^\tau = D_i^t\}$ is the time at which we last trained the model for shard $i$ in the unlearning algorithm. $\qquad\square$

**Theorem 5.1** (Unlearning Guarantees). *If for every round $t$, the sequence of publishing functions $\{f_{publish}^{t'}\}_{t' \leq t}$ is $(\epsilon, \delta)$-differentially private in the random seeds $r \sim \mathcal{P}^k$ of the algorithms for $0 < \epsilon \leq 1/2$ and $0 < \delta < \epsilon$, then $\mathcal{R}_{\mathcal{A}^{\text{distr}}}$ is an $(\alpha, \beta, \gamma)$-unlearning algorithm for $\mathcal{A}^{\text{distr}}$ where*

$$\alpha = O\left(\epsilon^2 k + k\sqrt{\delta/\epsilon}\right), \quad \beta = \gamma = O\left(\sqrt{e^{-\epsilon^2 k} + k\sqrt{\delta/\epsilon}}\right)$$

*Proof.* Lemma 5.1 provides that $\mathcal{R}_{\mathcal{A}^{\text{distr}}}$ is a $(0, 0, 0)$-unlearning algorithm for $\mathcal{A}^{\text{distr}}$ against any nonadaptive update requester.

Note that because the randomness used in each shard $i \in [k]$ is always independent and there is a symmetry across these shards in both $\mathcal{A}^{\text{distr}}$ and $\mathcal{R}_{\mathcal{A}^{\text{distr}}}^{\text{iter}}$, we can imagine drawing all the randomness required for each shard throughout the interaction prior to the interaction $r \sim \mathcal{P}^k$ such that each shard $i \in [k]$ relies $r_i$ on as the source of its randomness.

Now, note that the state kept by $\mathcal{R}_{\mathcal{A}}^{\text{distr}}$ consists of the shards $\{D_i^{t-1}\}_i$ and the models trained via $\mathcal{A}^{\text{single}}$ on those shards $\{\theta_i^{t-1}\}_i$. Hence, at any round $t$, given access to initial dataset $D^0$, previous update requests $u^{\leq t-1}$, and the randomness that has been drawn prior to the interaction $r$, we can deterministically determine the state $s^{t-1} = (\{D_i^{t-1}\}_i, \{\theta_i^{t-1}\}_i)$, meaning there exists some deterministic mapping $g^{t-1}$ such that $s^{t-1} = g^{t-1}(D^0, u^{\leq t-1}, r)$.

Therefore, we can combine the $(0, 0, 0)$-deletion guarantee promised by Lemma 5.1 with Theorem 4.1 to conclude that $\mathcal{R}_{\mathcal{A}}^{\text{distr}}$ must be $(\alpha, \beta, \gamma)$-unlearning algorithm for $\mathcal{A}^{\text{distr}}$. $\qquad\square$

**Theorem 5.2** (Run-time Guarantees). *Let $p = 1/k$. Suppose the publishing functions satisfy the differential privacy requirement of Theorem 5.1. Let $N^t$ denote the number of times $\mathcal{R}_{\mathcal{A}}^{\text{distr}}$ calls $\mathcal{A}^{\text{single}}$ at round $t$. We have that $N^0 = k$, and for every round $t \geq 1$: 1) if the update requester is non-adaptive, for every $\xi$, with probability at least $1 - \xi$, $N^t \leq 1 + \sqrt{2\log(1/\xi)}$. 2) if the update requester is adaptive, for every $\xi$, with probability at least $1 - \xi$, $N^t \leq 1 + \sqrt{2\log((n + t)/\xi)}$. Furthermore, for $\xi > \delta'$, with probability at least $1 - \xi$, we have*

$$N^t \leq 1 + \min\left\{\sqrt{2\log(2(n + t)/(\xi - \delta'))}, \sqrt{2\epsilon' + 2\log(2/(\xi - \delta'))}\right\}$$

*where $\epsilon' = O\left(\epsilon^2 k + k\sqrt{\delta/\epsilon}\right)$ and $\delta' = e^{-\epsilon^2 k} + O\left(k\sqrt{\delta/\epsilon}\right)$*

*Proof.* Throughout we use $Bin(k, p)$ to denote a binomial random variable with parameters $k$ (number of trials) and $p$ (success probability). First we state the following fact:

**Fact B.1** (Binomial Tail Bound). *Let $X \sim Bin(k,p)$ and let $\mu := kp$. We have that for every $\eta \geq 0$,*

$$\Pr\left[X \geq (1+\eta)\mu\right] \leq e^{-\frac{\eta^2 \mu}{2+\eta}}$$

*which in turn implies, for every $\delta$, with probability at least $1 - \delta$,*

$$X \leq \left(1 + \frac{\sqrt{\log^2\left(1/\delta\right) + 8\mu \log\left(1/\delta\right)} - \log\left(1/\delta\right)}{2\mu}\right)\mu \leq \mu + \sqrt{2\mu \log\left(1/\delta\right)}$$

Fix any round $t \geq 1$ of the update, and let $\mu = kp$ throughout. Suppose the update requester is non-adaptive. If the update of round $t$ is an addition, then $N^t \sim Bin(k,p)$ by construction. If the update of round $t$ is a deletion: $u^t = (z^t, \text{'delete'})$, then

$$N^t = \sum_{i=1}^{k} \mathbb{1}\left[z^t \in D_i^{t-1}\right]$$

But the update requester being non-adaptive (implying $z^t$ is independent of the randomness of the algorithms), together with Lemma B.1, imply that $N^t$ is a sum of *independent* Bernoulli random variables with parameter $p$; hence, $N^t \sim Bin(k,p)$. Therefore, if the update requester is non-adaptive, we can apply Fact B.1 to conclude that for every $\xi$, with probability at least $1 - \xi$, we have

$$N^t \leq \mu + \sqrt{2\mu \log\left(1/\xi\right)}$$

which proves the first part of the theorem for the choice of $p = 1/k$. Now suppose the update requester is adaptive. If the update of round $t$ is an addition, then $N^t \sim Bin(k,p)$ by construction, and therefore using Fact B.1, with probability at least $1 - \xi$, we have $N^t \leq \mu + \sqrt{2\mu \log\left(1/\xi\right)}$. Now suppose the update is a deletion: $u^t = (z^t, \text{'delete'})$. We have in this case that

$$N^t = \sum_{i=1}^{k} \mathbb{1}\left[z^t \in D_i^{t-1}\right]$$

First note that we have the following upper bound

$$N^t \leq \sup_{z \in D^{t-1}} \sum_{i=1}^{k} \mathbb{1}\left[z \in D_i^{t-1}\right] \leq \sup_{z \in D^0 \cup \{z^1, \ldots, z^{t-1}\}} \sum_{i=1}^{k} \mathbb{1}\left[z \in D_i^{t-1}\right] \tag{5}$$

where $\{z^1, \ldots, z^{t-1}\}$ are the data points that have been requested to be added or deleted by the update requester in the previous rounds. Here, in the worst case (to get upper bounds), we are assuming that all previous $t - 1$ updates are addition requests. Note that for every $z \in D^0 \cup \{z^1, \ldots, z^{t-1}\}$, the number of shards that contain $z$ is an independent draw from a $Bin(k,p)$ distribution, by construction. We therefore have that

$$\sup_{z \in D^0 \cup \{z^1, \ldots, z^{t-1}\}} \sum_{i=1}^{k} \mathbb{1}\left[z \in D_i^{t-1}\right] \overset{d}{=} \sup_{1 \leq j \leq n+t-1} X_j \tag{6}$$

where the equality is in distribution, and $X_j \sim Bin(k,p)$. Now, combining Equations (5) and (6), and using Fact B.1, we get that for every $\eta \geq 0$,

$$\Pr\left[N^t \geq (1+\eta)\mu\right] \leq \sum_{j=1}^{n+t-1} \Pr\left[X_j \geq (1+\eta)\mu\right] \leq (n+t)e^{-\frac{\eta^2 \mu}{2+\eta}}$$

which implies, for every $\xi \geq 0$, with probability at least $1 - \xi$,

$$N^t \leq \mu + \sqrt{2\mu \log\left((n+t)/\xi\right)}. \tag{7}$$

We will prove another upper bound using the max-information bound. Recall that our distributed algorithms can be seen as drawing all the randomness $r \sim \mathcal{P}^k$ upfront for some distribution $\mathcal{P}$ (one draw from $\mathcal{P}$ per shard). Since the update sequence $u^{\leq t}$ (which is a post processing of the published

objects) is guaranteed to be $(\epsilon, \delta)$-differentially private in $r$, we get using the max-information bound that, for every $\eta \geq 0$,

$$\Pr\left[N^t \geq (1+\eta)\mu\right] \leq e^{\epsilon'} \Pr_{(r \otimes u^{\leq t})}\left[N^t \geq (1+\eta)\mu\right] + \delta' \qquad (8)$$

where on the left hand side the probability is taken with respect to the joint distribution of $r$ and $u^{\leq t}$, and on the right hand side $(r \otimes u^{\leq t})$ means $r$ and $u^{\leq t}$ are drawn independently from their corresponding marginal distributions. But when $r$ and $u^{\leq t}$ are drawn independently (i.e., the update requester is non-adaptive), $N^t \sim Bin(k, p)$ as we have shown in the first part of this theorem.

$$\Pr_{(r \otimes u^t)}\left[N^t \geq (1+\eta)\mu\right] = \Pr\left[Bin(k, p) \geq (1+\eta)\mu\right] \leq e^{-\frac{\eta^2 \mu}{2+\eta}} \qquad (9)$$

Therefore, combining Equations (8) and (9), we get that

$$\Pr\left[N^t \geq (1+\eta)\mu\right] \leq e^{\epsilon' - \frac{\eta^2 \mu}{2+\eta}} + \delta'$$

which in turn implies, for every $\xi > \delta'$, with probability at least $1 - \xi$,

$$N^t \leq \mu + \sqrt{2\mu\left(\epsilon' + \log\left(1/(\xi - \delta')\right)\right)} \qquad (10)$$

Combining the bounds of Equations (7) and (10), we get that for every $\xi > \delta'$, with probability $1 - \xi$,

$$N^t \leq \mu + \min\left\{\sqrt{2\mu\log\left(2(n+t)/(\xi - \delta')\right)}, \sqrt{2\mu\left(\epsilon' + 2\log\left(2/(\xi - \delta')\right)\right)}\right\}$$

which completes the proof by the choice of $p = 1/k$ ($\mu = kp = 1$). $\qquad \square$

---

**Algorithm 4:** `PrivatePredictionInteraction`$(\epsilon', \epsilon, \delta, k)$

---

$l = 0$
**for** $t = 1, \dots, T$ **do**
$\quad$ **if** $l > \left\lfloor \frac{\epsilon^2}{8(\epsilon')^2 \ln(\frac{1}{\delta})} \right\rfloor$ // "Restart" $\mathcal{R}_{\mathcal{A}}^{\texttt{distr}}$ when privacy budget is exhausted
$\quad$ **then**
$\quad\quad D_i^t = \texttt{Sampler}_i(D^t, p)$ and $\theta_i^t = \mathcal{A}_i^{\texttt{single}}(D_i^t)$ for each $i \in [k]$
$\quad\quad$ Update $s^t = (\{D_i^t\}_i, \{\theta_i^t\}_i)$
$\quad\quad l = 0$
$\quad$ **else**
$\quad\quad \{\theta_i^t\}_i = \mathcal{R}_{\mathcal{A}^{\texttt{distr}}}(D^{t-1}, u^t, s^{t-1})$
$\quad$ **while** there is a prediction request for some $x$ **do**
$\quad\quad$ Publish $\hat{y} = \texttt{PrivatePredict}_{\epsilon'}^k(\{\theta_i^t\}_i, x)$
$\quad\quad l = l + 1$

---

**Lemma B.2.** *Assume $\epsilon < 1$ and $\delta > 0$. Then, $(\hat{y}_1, \dots, \hat{y}_l)$ is $(\epsilon, \delta)$-differentially private in $\{\theta_i\}_i$ where $\hat{y}_j = \texttt{PrivatePredict}_{\epsilon'}^k(\{\theta_i\}, x_j)$ and*

$$l = \left\lfloor \frac{\epsilon^2}{8(\epsilon')^2 \ln(\frac{1}{\delta})} \right\rfloor.$$

*Proof.* This claim holds immediately by the $(\epsilon', 0)$-differential privacy of `PrivatePredict`$_{\epsilon'}^k$ and the advanced composition theorem. See Corollary 3.21 in Dwork and Roth [2014] for details. $\qquad \square$

**Theorem 5.3.** *The models $\{\{\theta_i^t\}_i\}_t$ in `PrivatePredictionInteraction`$(\epsilon', \epsilon, \delta, k)$ satisfy $(\alpha, \beta, \gamma)$-unlearning guarantee for $\mathcal{A}^{distr}$ where $\alpha = O\left(\epsilon^2 k + k\sqrt{\delta/\epsilon}\right)$ and $\beta, \gamma = O\left(\sqrt{e^{-\epsilon^2 k} + k\sqrt{\delta/\epsilon}}\right)$, if $0 < \epsilon \leq 1/2$ and $0 < \delta < \epsilon$.*

**Algorithm 5:** $\mathcal{A}^{\text{SISA}}$: Learning Algorithm for SISA

---

*Proof.*    **Input**: dataset $D \equiv D^0$ of size $n$

 Draw the shards: $D^0_{i \in [k]} = \texttt{RandomAssignPartition}(D^0, k)$.

 Train the models: $\theta^0_{i \in [k]} = \mathcal{A}^{\text{single}}(D^0_i)$, for every $i \in [k]$.

 Save the state: $s^0 = (\{D^0_i\}_{i \in [k]}, \{\theta^0_i\}_{i \in [k]})$

 **Output**: $\{\theta^0_i\}_{i \in [k]}$

---

**Algorithm 6:** $\mathcal{R}_{\mathcal{A}^{\text{SISA}}}$: Unlearning Algorithm for SISA: $t$'th round of unlearning

---

 **Input**: dataset $D^{t-1}$, update $u^t = (z^t, \bullet^t)$, state $s^{t-1} = (\{D^{t-1}_i\}_{i \in [k]}, \{\theta^{t-1}_i\}_{i \in [k]})$

 **if** $\bullet^t = {}'\texttt{delete}'$ **then**

  $i = j \in [k]$, where $z^t \in D^{t-1}_j$

 **else**

  $i = \texttt{randint}(1, 2, \ldots, k)$

 Update the shards: $D^t_i = \begin{cases} D^{t-1}_j \circ u^t & \text{if } i = j \\ D^{t-1}_j & \text{otherwise} \end{cases}$, for every $j \in [k]$.

 Update the models: $\theta^t_j = \begin{cases} \mathcal{A}^{\text{single}}(D^t_j) & \text{if } i = j \\ \theta^{t-1}_j & \text{otherwise} \end{cases}$, for every $i \in [k]$.

 Update the state: $s^t = (\{D^t_i\}_{i \in [k]}, \{\theta^t_i\}_{j \in [k]})$

 **Output**: $\{\theta^t_i\}_{i \in [k]}$

---

*Proof.* Suppose full retraining occurs in rounds $(t_1, t_2, \ldots, t_G)$ where we always have $t_1 = 0$ and $l > \left\lfloor \frac{\epsilon^2}{8(\epsilon')^2 \ln(\frac{1}{\delta})} \right\rfloor$ at round $t_g$ for any $g > 1$.

At any round $t_g$ when full retraining occurs, we can imagine restarting $\mathcal{R}^{\text{distr}}_{\mathcal{A}}$ by resetting the internal round as $t = 0$ and drawing fresh randomness $r \sim \mathcal{P}^k$, which determines the new initial state $s^0$. Therefore, for any $g \in [G-1]$ and $t_g \le t < t_{g+1}$, we must have that $\{f^{t'}_{\text{publish}}\}_{t_g \le t' \le t}$ are $(\epsilon, \delta)$-differentially private in the randomness $r$ drawn in round $t_g$. Then, we can appeal to Theorem 5.1 to conclude that for any $g \in [G-1]$ and $t_g \le t < t_{g+1}$, we have

$$\forall E \subseteq \Theta^*: \quad \Pr\left[\{\theta^t_i\}_i \in E \mid (u_{t_g}, \ldots, u_t)\right] \le e^\alpha \cdot \Pr\left[\mathcal{A}\left(D^t\right) \in E\right] + \beta.$$

Because we are redrawing fresh randomness $r \sim \mathcal{P}^k$ at $t_g$, we can combine combine all the previous unlearning guarantees in the previous $(t_{g'-1}, t_{g'})$ for $g' < g$ to conclude that at any round $t \in [T]$

$$\forall E \subseteq \Theta^*: \quad \Pr\left[\{\theta^t_i\}_i \in E \mid u^{\le t}\right] \le e^\alpha \cdot \Pr\left[\mathcal{A}\left(D^t\right) \in E\right] + \beta.$$

$\square$

## C    Details From Section 3

### C.1    Proof of Theorem 3.1

**Theorem 3.1.** *There are learning and unlearning algorithms in the SISA framework $(\mathcal{A}, \mathcal{R}_{\mathcal{A}})$ such that for any $\alpha$, and any $\beta, \gamma < 1/4$, $\mathcal{R}_{\mathcal{A}}$ is not an $(\alpha, \beta, \gamma)$-unlearning algorithm for $\mathcal{A}$.*

Define $\mathcal{A}^{\text{SISA}}$ and $\mathcal{R}_{\mathcal{A}^{\text{SISA}}}$ as Algorithms 5 and 6 respectively instantiated with the "lookup table" model $\mathcal{A}^{\text{single}}(D) = D$ and "lookup table" prediction rule $f_\theta$. In Algorithm 5, $\texttt{RandomAssignPartition}(D, k)$ assigns every $(x, y) \in D$ to one of the $k$ partitions uniformly at random. The prediction rule, given parameter $\theta = D$ and query point $x$, outputs $y$ if $(x, y) \in \theta$ and $\perp$ otherwise:

$$f_\theta(x) = \begin{cases} y & \text{if } (x, y) \in \theta, \\ \perp & \text{otherwise.} \end{cases}$$

We wish to show that there exists a dataset $D^0$ and adaptive update requester `UpdReq` such that for some update step $t \geq 1$, with probability at least $1 - \gamma$ over the draw of the update sequence $u^{\leq t} = (u^1, \ldots, u^t)$ from `UpdReq`, $\exists E \subseteq \Theta^* : \quad \Pr\left[\mathcal{R}_\mathcal{A}\left(D^{t-1}, u^t, s^{t-1}\right) \in E \,\middle|\, u^{\leq t}\right] > e^\alpha \cdot \Pr\left[\mathcal{A}\left(D^t\right) \in E\right] + \beta$. We prove this with the following example, instantiated for $k = 3$.

Consider dataset $D^0$ consisting of training examples $\{(x_i, y_i)\}_{i \in [2n]}$, $n \in \mathbb{Z}^+$ such that $D^0$ contains 2 copies each of $n$ distinct feature vectors $x$. Both copies of each distinct feature vector $x$ are paired with the same (arbitrary) label $y$.

Further, given ensemble model parameters $\{\theta_i\}_{i \in [k]} = \mathcal{A}^{\mathrm{distr}}(D)$, let the ensemble output the mode of the predictions made by the underlying models:

$$\hat{y}_i = \texttt{Mode}\left(\left\{f_{\theta_j}(x_i)\right\}_{j \in [k]}\right).$$

Let $\psi^0$, the published object after initial training, be the ensemble's predictions for each training point: $\psi^0 = f_{\mathrm{publish}}^0 = (\hat{y}_1^0, \hat{y}_2^0, \ldots, \hat{y}_{2n}^0)$.

Given these predictions, let $I = \{i_1, i_2, \ldots, i_t\} \subseteq [2n]$ be the indices for the points which were classified correctly. That is, $\forall i \in [2n] : i \in I$ if $y_i = \hat{y}_i$. Given $\psi^0$, let `UpdReq` be a function which outputs the deletion sequence $(u^1, u^2, \ldots, u^t)$ where each update request is responsible for deleting one of the correctly predicted points: $\forall j \in [t] : u^j = ((x_{i_j}, y_{i_j}), '\texttt{delete}')$.

Recall that our model is parameterized by a set of model parameters $\{\theta_i\}_{i \in [k]}$ and each $\theta_i$ is the dataset that shard is trained on. We now define the event $E$ of interest: the set of all models such that the ensemble attains zero accuracy on the remaining points $D^t = D^0 \circ (u^1, u^2, \ldots, u^t)$, which happens if and only if all identical points (both copies of the same point) fall into the same shard.

$$E^t = \left\{\{\theta_i\}_{i \in [k]} \text{ where } |\{\theta_i : (x, y) \in \theta_i\}| = 1 \text{ for all } (x, y) \in D^t\right\}$$

To make our final assertion, first note that $\Pr[\mathcal{R}_{\mathcal{A}^{\mathrm{SISA}}}(D^{t-1}, u^t, s^{t-1}) \in E | u^{\leq t}] = 1$ as `UpdReq` has requested all the correctly classified points to be deleted. We therefore need to show that

$$\Pr\left[\mathcal{R}_{\mathcal{A}^{\mathrm{SISA}}}\left(D^{t-1}, u^t, s^{t-1}\right) \in E \,\middle|\, u^{\leq t}\right] = 1 > e^\alpha \cdot \Pr\left[\mathcal{A}^{\mathrm{SISA}}\left(D^t\right) \in E\right] + \beta$$

equivalently, $\frac{1 - \beta}{e^\alpha} > \Pr\left[\mathcal{A}^{\mathrm{SISA}}\left(D^t\right) \in E\right]$ with probability $1 - \gamma$ over the randomness of the update sequence (which in this case is simply the randomness of the initial partition).

Note that $t$, the number of copies of points that were initially classified correctly is distributed as $\texttt{Binomial}(n, \frac{2}{3})$ because for each pair of identical $(x, y) \in D^0$, the probability that they fall in different shards initially is exactly $2/3$. Also, note that for any fixed $t \leq n - 1$,

$$\Pr[\mathcal{A}^{\mathrm{distr}}(D^t) \in E] = \frac{1}{3^{n-t}}.$$

Using the tail bound for the Binomial distribution (Fact B.1), we have that with probability $1 - \gamma$,

$$t \leq \frac{2n}{3} + \sqrt{\frac{4n}{3} \log\left(\frac{1}{\gamma}\right)}.$$

When $n \geq 13 \log(1/\gamma)$, we have $\frac{2n}{3} + \sqrt{\frac{4n}{3} \log\left(\frac{1}{\gamma}\right)} \leq 0.99n$. Hence, for sufficiently large $n$, we can conclude that with probability $1 - \gamma$,

$$\Pr[\mathcal{A}^{\mathrm{distr}}(D^t) \in E] \leq \frac{1}{3^{0.01n}}.$$

Finally, for any $c = \frac{1-\beta}{e^\alpha} > 0$, there exists a $D^0$ such that $c > \Pr\left[\mathcal{A}^{\mathrm{SISA}}\left(D^t\right) \in E\right]$ with probability $1 - \gamma$ because we can choose a sufficiently large $n$ such that $n \geq 13 \log(1/\gamma)$ and $\frac{1}{3^{0.01n}} \leq c$, i.e., we can choose:

$$n \geq \max\left\{13 \log(1/\gamma), \frac{100 \log(1/c)}{\log 3}\right\}$$

$\square$

## C.2  Failures in (0, 0, 0)-Unlearning Beyond Section 3

Observable failures in unlearning guarantees for algorithms in the SISA framework go beyond the simplistic setting constructed in Section 3. In this section, we describe a more natural setting in which we employ the learning and unlearning algorithms for SISA $(\mathcal{A}, \mathcal{R}_{\mathcal{A}})$ and are able to construct an adaptive deletion sequence (only given discrete predictions through $f_{\text{publish}}$) which, to a high degree of confidence, rejects the null hypothesis that $(\mathcal{A}, \mathcal{R}_{\mathcal{A}})$ satisfy a perfect $(0, 0, 0)$-unlearning guarantee.

In Section 3 we explicitly define a base model $f_\theta$ which relies on the fact that each point is copied twice to reveal perfect information about how points were partitioned through its predictions. Here, we define a new model which relaxes this condition. Given a query point $x$, rather than return the label of an exactly matching point, the model $f_{D,\tau}(x)$ is additionally parameterized by a threshold $\tau$. This model, reminiscent of 1-nearest neighbors, returns the label of the closest point $(x', y') \in D$ where $|x - x'|_2 \leq \tau$, and $\perp$ otherwise, essentially treating nearby points as "identical."

Here we define $\mathcal{A}^{\text{SISA}}$ and $\mathcal{R}_{\mathcal{A}^{\text{SISA}}}$ as Algorithms 5 and 6 respectively, instantiated with $\mathcal{A}^{\text{single}}(D) = D$ and prediction rule $f_{D,\tau}$. We assume the null hypothesis that $\mathcal{A}$ and $\mathcal{R}_{\mathcal{A}}$ satisfy a $(0, 0, 0)$-unlearning guarantee.

To make an assertion about this hypothesis, we train an ensemble using three shards as before. We then execute a similar experiment to that as described in Section 3 in which, after initial training, we publish the aggregated discrete predictions for each training point and delete a random subset of correctly classified points. We then observe the accuracy of the ensemble on the remaining training points. Our hypothesis, the same as before, is that the resulting accuracy will be lower in the adaptive deletion setting than the retrain setting with high probability.

We then define an event $E$ of interest to be when the training accuracy after the adaptive deletion sequence falls below a cutoff $c \in [0\%, 100\%]$ after deleting all correctly classified points. We can then estimate the probability of this event by defining an indicator for each trial which is 1 if the training accuracy falls below this threshold and 0 otherwise. We then run many trials to calculate confidence intervals on our estimate of this probability under either setting. If the confidence intervals are non-overlapping at some confidence level, we can then reject the null hypothesis at some level of confidence.

Our concrete experiment samples 1,000 random points from MNIST, each being either a "0" or "1" (preprocessing each image by dividing each pixel value by 255). With $\tau = 6.5$, this setting is "plausible" in the sense that this model's performance on held-out test data is nontrivial (approximately $91.2\%$ test accuracy before deletion) for a common benchmark task. We then delete $t$ points (a uniformly random subset of correctly predicted points), observe the average accuracies across trials on remaining points under the adaptive setting and the retrain scenario. We grid search for the $c$ which yields the largest difference in the confidence intervals (since the unlearning guarantee should hold for all $c$). Under these conditions we find that after 200 trials, we attain $97.5\%$ confidence intervals on our statistic to be those shown in Figure 2. We see that for deletion sequences of 200 points or more we can induce a reliable difference in this statistic at a high level of confidence, rejecting the null hypothesis at $p \leq 0.05$ that $\mathcal{A}^{\text{SISA}}$ and $\mathcal{R}_{\mathcal{A}^{\text{SISA}}}$ satisfy a perfect $(0, 0, 0)$-unlearning guarantee.

# D  Full Experimental Details From Section 6

Choices in hyperparameters and and model architecture for experiments presented in Section 6 were inspired by those used by Papernot et al. [2021]. All models were optimized using momentum with mass equal to 0.9. The clipping parameter (upper bound on maximum $\ell_2$-norm of per-example gradients) used in DP-SGD for all experiments was equal to 0.1. For certain experiments, the batch size was reduced from what was presented in Papernot et al. [2021] to reduce computational cost. Each experiment was repeated with new random seeds across 300 trials to get the confidence intervals displayed in Figure 1. The precise model definition for each experiment is given below:

```
Sequential(
  Conv(out_chan=16, filter_shape=(8, 8), padding='SAME', strides=(2, 2)),
  Tanh,
  MaxPool(window_shape=(2, 2), strides=(1, 1)),
  Conv(out_chan=32, filter_shape=(4, 4), padding='VALID', strides=(2, 2)),
```

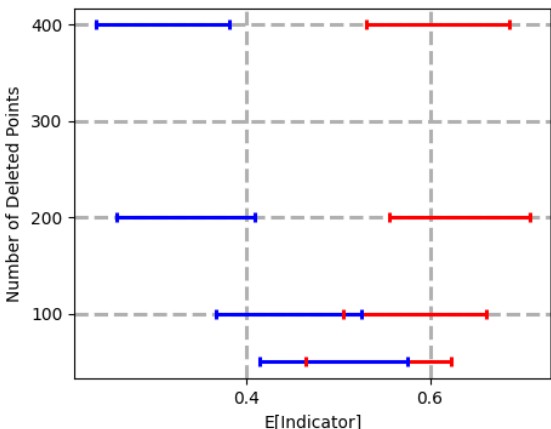

Figure 2: Confidence intervals for the indicator defined in Section C.2 as a function of the number of deleted points. Red confidence intervals correspond to the statistic after the adaptive deletion sequence, and blue confidence intervals correspond to the statistic after full retraining. For deletion sequences of 200 points or more we can induce a reliable enough difference in the confidence intervals to reject the null hypothesis at $p \leq 0.05$ that $\mathcal{A}^{\text{SISA}}$ and $\mathcal{R}_{\mathcal{A}^{\text{SISA}}}$ satisfy a perfect $(0, 0, 0)$-unlearning guarantee in a realistic setting.

```
  Tanh,
  MaxPool(window_shape=(2, 2), strides=(1, 1)),
  Flatten,
  Dense(out_dim=32),
  Tanh,
  Dense(out_dim=num_classes)
)
```

In our experiments we make use of 3 common benchmark machine learning datasets. The MNIST database of handwritten digits given by Lecun et al. [1998] consists of 70,000 $28 \times 28$ images of handwritten digits, each belonging to one of 10 classes characterizing the digit shown in each image. MNIST is made available under the Creative Commons Attribution-Share Alike 3.0 license. The Fashion-MNIST dataset given by Xiao et al. [2017] consists of 70,000 $28 \times 28$ grayscale images of pieces of clothing, each belonging to one of 10 classes (e.g. t-shirt, dress, sneaker, etc.) Fashion-MNIST is made available under the MIT license. The CIFAR-10 dataset given by Krizhevsky and Hinton [2009] consists of 60,000 $32 \times 32$ images in RGB format, each belonging to one of 10 classes characterizing the class of the object given in each image (e.g. airplane, automobile, bird, etc.) CIFAR-10 is made available under the MIT license.

With respect to computing environment, experiments were conducted using the JAX deep learning framework developed by Bradbury et al. [2018]. Experiments were run using 1 Tesla V100 GPU using CUDA version 11.0, where an individual trial (training a full ensemble, deleting targeted points, and retraining) would take approximately 1-6 minutes depending on the number of shards, iterations, image size, etc.

| $\mathbb{E}$[Indicator] | Acc. (after) | Acc. (before) | Noise mult. | Shard pred. acc. |
|---|---|---|---|---|
| **CIFAR-10$_{k=6}$** | | | | |
| $[0.890, 0.952]$ | $0.507 \pm 0.025$ | $0.572 \pm 0.011$ | $0$ | $0.303 \pm 0.005$ |
| $[0.784, 0.869]$ | $0.504 \pm 0.020$ | $0.554 \pm 0.010$ | $0.15$ | $0.257 \pm 0.004$ |
| $[0.670, 0.772]$ | $0.487 \pm 0.021$ | $0.525 \pm 0.011$ | $0.22$ | $0.229 \pm 0.003$ |
| $[0.490, 0.603]$ | $0.455 \pm 0.025$ | $0.484 \pm 0.012$ | $0.3$ | $0.205 \pm 0.003$ |
| **CIFAR-10$_{k=2}$** | | | | |
| $[0.947, 0.987]$ | $0.448 \pm 0.033$ | $0.521 \pm 0.019$ | $0$ | $0.655 \pm 0.012$ |
| $[0.797, 0.881]$ | $0.433 \pm 0.026$ | $0.475 \pm 0.015$ | $0.2$ | $0.587 \pm 0.007$ |
| $[0.638, 0.744]$ | $0.419 \pm 0.025$ | $0.452 \pm 0.015$ | $0.25$ | $0.567 \pm 0.006$ |
| $[0.493, 0.607]$ | $0.399 \pm 0.027$ | $0.427 \pm 0.015$ | $0.3$ | $0.550 \pm 0.005$ |
| **Fashion-MNIST$_{k=6}$** | | | | |
| $[0.819, 0.899]$ | $0.849 \pm 0.011$ | $0.874 \pm 0.004$ | $0$ | $0.248 \pm 0.007$ |
| $[0.662, 0.765]$ | $0.838 \pm 0.011$ | $0.854 \pm 0.005$ | $0.4$ | $0.215 \pm 0.005$ |
| $[0.540, 0.652]$ | $0.823 \pm 0.009$ | $0.834 \pm 0.006$ | $0.6$ | $0.198 \pm 0.004$ |
| $[0.477, 0.590]$ | $0.810 \pm 0.012$ | $0.820 \pm 0.006$ | $0.75$ | $0.190 \pm 0.004$ |
| **Fashion-MNIST$_{k=2}$** | | | | |
| $[0.976, 0.999]$ | $0.826 \pm 0.016$ | $0.863 \pm 0.006$ | $0$ | $0.597 \pm 0.014$ |
| $[0.797, 0.881]$ | $0.808 \pm 0.013$ | $0.828 \pm 0.007$ | $0.5$ | $0.555 \pm 0.010$ |
| $[0.607, 0.715]$ | $0.791 \pm 0.016$ | $0.807 \pm 0.008$ | $0.7$ | $0.538 \pm 0.007$ |
| $[0.497, 0.610]$ | $0.763 \pm 0.020$ | $0.781 \pm 0.009$ | $1$ | $0.523 \pm 0.005$ |
| **MNIST$_{k=6}$** | | | | |
| $[0.849, 0.922]$ | $0.973 \pm 0.004$ | $0.978 \pm 0.002$ | $0$ | $0.201 \pm 0.004$ |
| $[0.729, 0.824]$ | $0.965 \pm 0.005$ | $0.969 \pm 0.003$ | $0.4$ | $0.186 \pm 0.003$ |
| $[0.583, 0.694]$ | $0.940 \pm 0.009$ | $0.945 \pm 0.004$ | $0.8$ | $0.178 \pm 0.003$ |
| $[0.493, 0.607]$ | $0.913 \pm 0.018$ | $0.923 \pm 0.007$ | $1.1$ | $0.176 \pm 0.003$ |
| **MNIST$_{k=2}$** | | | | |
| $[0.927, 0.976]$ | $0.962 \pm 0.007$ | $0.971 \pm 0.003$ | $0$ | $0.540 \pm 0.006$ |
| $[0.769, 0.857]$ | $0.959 \pm 0.008$ | $0.968 \pm 0.003$ | $0.8$ | $0.534 \pm 0.006$ |
| $[0.587, 0.697]$ | $0.953 \pm 0.007$ | $0.962 \pm 0.004$ | $1.3$ | $0.530 \pm 0.006$ |
| $[0.497, 0.610]$ | $0.949 \pm 0.008$ | $0.957 \pm 0.004$ | $1.6$ | $0.527 \pm 0.006$ |

Table 1: Numerical representation of results displayed in Figure 1. The $x$ axis in Figure 1 corresponds to column "$\mathbb{E}$[Indicator]", and the $y$ axis corresponds to column "Acc. (after)". Column "$\mathbb{E}$[Indicator]" represents the 95% confidence interval of the indicator after 300 trials. Columns "Acc. (before)" and "Acc. (after)" represent the accuracy of the ensemble on a held-out test set (5,000 points each) before and after deleting approximately half of the points from the ensemble, with confidence intervals given by two standard deviations above and below the observed mean. "Noise multiplier" represents the standard deviation of Gaussian noise applied to each per-example gradient during DP-SGD. Shard prediction accuracy denotes the prediction accuracy of the adversary in targeting models when deleting points, where random guessing would achieve an accuracy of $1/(\text{\# shards})$.

| Experiment | Points per shard | Batch Size | Iterations | Step size |
|---|---|---|---|---|
| CIFAR-10$_{k=6}$ | 8000 | 64 | 4000 | 1.0 |
| CIFAR-10$_{k=2}$ | 8000 | 64 | 4000 | 1.0 |
| Fashion-MNIST$_{k=6}$ | 6000 | 256 | 1500 | 4.0 |
| Fashion-MNIST$_{k=2}$ | 6000 | 256 | 2000 | 4.0 |
| MNIST$_{k=6}$ | 6000 | 64 | 2500 | 0.5 |
| MNIST$_{k=2}$ | 6000 | 256 | 2000 | 0.5 |

Table 2: Remaining hyperparameter settings for each experiment, by dataset.