# OpenReview forum: "Adaptive Machine Unlearning"
_NeurIPS.cc/2021/Conference — NeurIPS 2021 Poster_

### Official Review · Reviewer_QxA8 · 2021-07-07

**Rating:** 7
**Confidence:** 3

**Summary:**

The paper highlights an important problem in machine unlearning in which privacy can be lost between deletion requests if the deletion requests are chosen based on the published intermediate models (e.g. model predictions, the models themselves, etc.), even if the unlearning mechanism guarantees exact deletion for non-adaptive deletion sequences (e.g. the SISA framework).

The paper then shows that if an unlearning approach provides guarantees against non-adaptive deletion sequences, and differential privacy can be used to obfuscate the internal randomness of the intermediate models, then that unlearning approach satisfies data deletion guarantees against an adaptive sequence of deletions.

**Limitations And Societal Impact:**

Yes.

**Main Review:**

Strengths

As far as I know, this is original work on addressing an important problem of possible privacy leakage via the intermediate publishing of models between machine unlearning updates in response to deletion requests. It combines well-established methods of machine unlearning and differential privacy to provide a framework for developing new machine-unlearning algorithms with adaptive data deletion guarantees.

The theorems in the paper are supported by proofs in the supplemental materials that appear to be technically sound.

I think this is a relatively new and under-explored topic in the burgeoning field of machine unlearning, one that may be very important for any parties interested in adhering to legislation surrounding the "Right to be Forgotten."


Weaknesses

The paper is organized well and follows a logical order; however, I think the paper could benefit from some simple examples early in the paper to enhance the understandability of the topic, especially the difference between non-adaptive and adaptive deletion sequences.

I think there could be a clearer discussion or perhaps definition about how some currently existing machine unlearning approaches already satisfy adaptive deletion guarantees while others do not.

The work seems to be mainly based on the SISA unlearning framework; can this methodology be adapted to other machine unlearning approaches / frameworks?

Does the proposed approach of making a non-adaptive unlearner into an adaptive unlearner incur any deletion efficiency costs? Or does the cost mainly affect predictive performance?

The amount of added noise needed to reject the null hypothesis (Figure 1) actually seems to cause a significant degradation in predictive performance; most notably for Fashion-MNIST (k=2 and k=6) and MNIST (k=6).

Minor Weaknesses

Figure 1(f - bottom) has duplicate numbers on the y-axis.

How much privacy is lost when using an adaptive deletion sequence over a non-adaptive deletion sequence?

How would a machine-learning practitioner go about selecting values for the hyperparameters: alpha, beta, and gamma?


===========POST-REBUTTAL RESPONSE============

I thank the authors for their detailed and thorough response. I think the authors do a solid job of highlighting a new under-explored problem in machine unlearning, and propose a solution that adapts a popular method of machine unlearning to the adaptive deletion case; thus, I have increased my score to 7 (good paper; accept). In the next revision, I encourage the authors to provide clear illustrative examples of adaptive deletion early in the paper to more cleanly delineate between adaptive and non-adaptive deletion sequences. I also think a more thorough discussion that describes what existing unlearning approaches already satisfy the adaptive deletion case could be beneficial.

**Time Spent Reviewing:**

5.5

---

> ### Author Response · Authors · 2021-08-10
> **Response**
>
> Thank you for the careful review! To briefly answer your questions:
>
> *Examples* We agree that the paper begins at a high level of abstraction, which might be confusing. In the next version, we plan to move Section 5 (attacks on SISA) before our theoretical results, because this section makes very concrete what an adaptive deletion sequence can look like.
>
> *Other approaches* Our framework applies to any setting in which it is possible to protect the randomness of the internal state of the algorithm with differential privacy. This is possible in a very general setting: For example, whenever random bits affect the outcome only through their effect on single data points (as they do e.g. when used to select mini batches), then differential privacy of the data implies differential privacy of the randomness. We focus on the SISA setting in this paper because our main result is a _reduction_ to data deletion algorithms that have guarantees for non-adaptive deletion sequences, and SISA is the only such algorithm in the literature that has such guarantees for arbitrary non-convex models. However, we expect our approach to be applicable to new methods that are yet to be developed as well.
>
> *Deletion Efficiency Costs*
>
> To make the algorithm robust to adaptive data deletions, we add noise to the aggregation step of SISA. This does result in loss of classification accuracy, which we quantify experimentally in Section 5. It does not on its own result in any loss of deletion efficiency: if the data deletion sequence is non-adaptive, then we continue to only have to retrain one shard per iteration in expectation. Adaptive data deletion sequences can lead to higher retraining costs, but  we can bound these in Theorem 4.2.
>
> The difference between the non-adaptive and adaptive version is the addition of noise in order to hide the randomness used in the algorithm. So, there isn’t any privacy that is lost as a result of running the adaptive algorithm, but we do formalize the tradeoff between the amount of noise added and unlearning guarantee (Thm 3.1), running time (e.g. Thm 4.2), and the quality of the predictions.
>
> *Experimental Accuracy degradation*
> Note that the y-axis is scaled to promote readability, and so is zoomed in on a compressed region of the error space. So, for example, the accuracy decrease due to noise addition in Figure 1b (Fashion-MNIST k=6)  is from ~ 0.85 (in the no noise case) to ~ 0.81 (with sufficient noise to defeat our attack). Similarly, in Figure 1c (MNIST k=6), accuracy decreases from ~ 0.97 with no noise to ~ 0.91 with enough noise to defeat our attack. We view these decreases in error (0.03 and 0.06 respectively) to be relatively small.
>
>
>
> *How to choose deletion parameters*
> This is a good question; here we repeat our answer to reviewer bzCS.  There is a standard way to think about this in differential privacy, that carries over to our data deletion setting. Informally, in our setting, we can think of $\beta$ and $\gamma$ as being set to be extremely small (e.g. exponentially small in $n$, say), and so we can think of these as low probability ``failure’’ events. $\alpha$ controls how hard it is to distinguish the distribution over models resulting from full retraining from the distribution that results from deletion. One way to think about the hardness of a hypothesis testing problem is by the relationship between Type I and Type II errors. If we ignore evidence entirely, and randomly guess that the output is the result of full retraining with probability $p$, then our Type I and Type II error rates are $p$ and $1-p$ respectively. The unlearning guarantee promises that except with some failure probability governed by $\gamma$ and $\beta$, any hypothesis test that has Type I error $p$ has type II error at most $1-e^\alpha p$. Thus as $\alpha$ becomes small, it says that the advantage of any hypothesis test over random guessing becomes small. The user can then choose $\alpha$ by reasoning about what advantage over random guessing is tolerable.

---

### Official Review · Reviewer_bzCs · 2021-07-16

**Rating:** 7
**Confidence:** 3

**Summary:**

This paper studies data deletion algorithms in an adaptive updating scenario.

Firstly, the authors define (adaptive) ($\alpha$,$\beta$,$\gamma$)-unlearning guarantee as well as its non-adaptive version for data deletion algorithms. With the techniques borrowed from differential privacy, the authors prove that an algorithm with the non-adaptive unlearning guarantee can be reduced to an algorithm with the adaptive unlearning guarantee.

Secondly, the authors apply the theoretical results to SISA unlearning algorithm. Specifically, an updating algorithm for SISA is proposed, which satisfies the adaptive unlearning guarantee. The authors also give an example showing that there exists updating algorithms that do not satisfy the adaptive unlearning guarantee.

**Ethical Concerns:**

None.

**Limitations And Societal Impact:**

This paper does not have any potential negative societal impact.

**Main Review:**

This paper tackles a very important problem, unlearning problem, and the authors do a good job of solving it in the context of adaptive (un)learning. The authors perfectly apply adaptive differential privacy into the field of machine unlearning, and are the first that formalize and solve the adaptive unlearning problem.

I only have one question remained: can the authors estimate the $\alpha$, $\beta$, and $\gamma$ parameters in the adaptive unlearning guarantee?

**Time Spent Reviewing:**

5

---

> ### Author Response · Authors · 2021-08-10
> **Response**
>
> Thank you for the positive review!
>
> To answer your question: the parameters that are input to the algorithm are $\epsilon$ and $\delta$ (parameters used for private predictions) which in turn give unlearning guarantees for the overall algorithm, parameterized by $(\alpha, \beta, \gamma)$, and the relationship between $(\epsilon, \delta)$ and $(\alpha, \beta, \gamma)$ is given in both Theorem 4.1 and 4.3. This gives the user the ability to choose $\epsilon$ and $\delta$ once they have decided on their target unlearning parameters $(\alpha, \beta, \gamma)$. Re: how to think about choosing $\alpha, \beta, \gamma$: this is a similar question to how one should choose differential privacy parameters $(\epsilon,\delta)$ when one is interested in privacy. This is a generally interesting question --- here we describe a standard way to think about this in differential privacy, that carries over to our data deletion setting. Informally, in our setting, we can think of $\beta$ and $\gamma$ as being set to be extremely small (e.g. exponentially small in $n$, say), and so we can think of these as controlling low probability ``failure’’ events. $\alpha$ controls how hard it is to distinguish the distribution over models resulting from full retraining from the distribution that results from deletion, in the absence of a failure event. One way to think about the hardness of a hypothesis testing problem is by the relationship between Type I and Type II errors. If we ignore evidence entirely, and randomly guess that the output is the result of full retraining with probability $p$, then our Type I and Type II error rates are $p$ and $1-p$ respectively. The unlearning guarantee promises that except with some failure probability governed by $\gamma$ and $\beta$, any hypothesis test that has Type I error $p$ has type II error at least $1-e^\alpha p$. Thus as $\alpha$ becomes small, it says that the advantage of any hypothesis test over random guessing becomes small. The user can then choose $\alpha$ by reasoning about what advantage over random guessing is tolerable.

---

### Official Review · Reviewer_bBvQ · 2021-07-17

**Rating:** 6
**Confidence:** 4

**Summary:**

The paper considers the task of machine unlearning with the modification that the deletion requests are adaptive i.e. they are a function of the random outputs (published model/predictions). The authors propose a generic reduction to convert oblivious unlearning methods to adaptive unlearning methods with small degradation in unlearning criteria.

**Limitations And Societal Impact:**

No direct societal impact.

**Main Review:**

Briefly, the machine unlearning problem is as follows: after model is trained on a dataset, there is request to delete a point from the training dataset. The goal is to design an efficient (faster than retraining) method such that its output is nearly indistinguishable from what we would have obtained had we trained on the dataset without the point.
The paper considers an interesting and important modification to the above setup: what if the choice of points to be deleted are *adaptive* i.e. they are selected as a function of the previous random output --  they motivate it by saying that users may choose to delete their data based on what the model reveals about them. While the problem of machine unlearning is in general important and has gained considerable attention recently, the modification is also well-motivated and interesting. One thing to note is that adaptivity considered in the paper is only with respect to *published* models and not the entire random state; however, this is reasonable too.

**General applicability of the reduction**: The paper makes a general sounding claim: a reduction to make (many?) oblivious unlearning methods adaptive. But I am not sure if there are many (known) methods where the reduction can be applied. The key idea is to make the published outputs differentially private - however the input (using which DP is defined) is not the dataset but $m$ i.i.d. bits used to generate the published output. Are there many methods which sample $m$ i.i.d values to generate the published state? It seems to me that the natural (and perhaps only?) candidates are aggregation/bagging based methods, where the each i.i.d. random variable corresponds to a bag. That authors instantiate their reduction using a known method SISA (Machine Unlearning, 2020. Bourtoule et al.), which is indeed of this nature. A few examples (other than aggregation) would help access the generality of the claim.

**No accuracy guarantees**: My major complaint is that, even though the authors present the result as a general reduction which hopefully can be applied to many existing methods, when they instantiate it with SISA , they provide no accuracy guarantees, even in say, convex learning settings. I don't know whether there exist no known accuracy guarantees for SISA, or the authors skipped this on purpose. But without accuracy guarantees, it is hard to evaluate if the proposed methods have overall non-trivial guarantees. From a theoretical viewpoint, I can simply take a constant function as my learning algorithm, it has has no accuracy guarantees, but it is trivial to unlearn, even with adaptive updates. I hope that the authors present rigorous accuracy analysis to identify regimes of accuracy where this method is provably *better* than deterministic retraining.

**Unlearning runtime guarantees**:
Since there are no accuracy guarantees, one avenue to access improvement is to consider the unlearning runtime of the proposed method against original (non-adaptive) method on adaptive edits.
If we do (non-adaptive) SISA training, then w.h.p. the worst data point can at most be in $\log n $ shards, so we retrain $\log n$ times and have **perfect** unlearning against adaptive updates.
Is the goal to shave this $\log n$ factor?
The authors proposal gives an approximate unlearning method (rather than perfect unlearning), and we replace this $\log n$ with a function of unlearning parameters. So, at best the result is a trade-off -- however, the choice of unlearning parameters with a *hand-wavy* accuracy analysis makes me doubt if this trade-off is good (see "Unlearning parameters" below)

**No $\log n$ and modified SISA**:
The authors consider SISA wherein they do sampling with replacing to construct the shards. Instead, if we do sampling without replacement, or if the dataset is i.i.d, as is typical in ML, we can just consider any deterministic partitioning. Then, unlearning time, with SISA, is constant, even for adaptive requests. In fact, in the original paper (Machine Unleraning, 2020), the authors say in Section B.1 "Sharding" -- *"By dividing the data into disjoint fragments and ..."*. Further discussion in this paragraph also points to a partitioning, even if randomized. Please answer if/how the above reasoning is incorrect, and if it is indeed correct, then why did the authors not mention this crucial change to the SISA method?

**Unlearning parameters**:
In differential privacy, it is standard to consider $\epsilon< $ "small constant" say two for reasonable privacy properties, and the same reasoning can be imported to the unlearning probabilities. In theorem 4.1, the parameter $\alpha = O(\epsilon^2 k)$ roughly, so for meaningful unlearning guarantees, we want $\epsilon = 1/\sqrt{k}$ -- this is a rather strong privacy guarantee.
Note that, if aggregation releases an averaged model, then the error due to privacy,
scales like $O(1/m\epsilon)$ where $m$ is the number of data points: here $m=k$, so error is $1/\sqrt{k}$. Thus, to get a good aggregation accuracy, we need large $k$, but then we would suffer in the error of each model in the shard since it is only trained on $n/k$ data points.
Perhaps we can do better if we only output private predictions than a model.
A formal accuracy analysis would explain these tradeoffs carefully and show if there is a setting with interesting guarantees.

**Confusion in experiment section**: The authors in line 342 say *`` .. adding various amounts of noise to the gradients in the model to satisfy finite levels of differential privacy (though much weaker privacy guarantees than would be needed to invoke our theorem)"*.
It seems that adding noise to gradients will produce a (standard) DP model output. However, the notion of DP to invoke their theorem is with respect to random bits  parametrizing the state, and is done by randomizing aggregation of predictions/model. Hence, I am confused by this comment, please explain.

**Missing reference to related work**:  A related paper Ullah et al. "Machine unlearning via algorithmic stability", 2021, is missing from related work.
I think (correct me if wrong) this work supports adaptive edits, perhaps with a (similar) logarithmic overhead in unlearning runtime.

**Time Spent Reviewing:**

6

---

> ### Author Response · Authors · 2021-08-10
> **Response**
>
> We thank you as well for your careful reading and the amount of time you obviously put into your review: it will improve the paper.
>
> We want to start off by clarifying what we think might be a crucial misunderstanding: *the problem of adaptivity does not merely increase the running time of previous approaches like SISA, it entirely invalidates their deletion guarantees!* So, in response to your question: _“If we do (non-adaptive) SISA training, then w.h.p. the worst data point can at most be in $\log n$  shards, so we retrain $\log n$  times and have perfect unlearning against adaptive updates. Is the goal to shave this $\log n$ factor?”_, the answer is *no*. It is true that adaptivity can increase the running time of SISA by at most a $\log n$ factor, but our main focus is that the adaptivity entirely destroys its deletion guarantees. Independent of run time, it becomes easy for an adversary to distinguish the (distribution over) models resulting from unlearning compared to those that result from full retraining. We demonstrate this in Section 5 --- so it is very much *not the case that it has perfect unlearning guarantees against adaptive updates*. This is the problem that we are able to fix via the main result of our paper; we will work to further clarify this in the exposition. Now we’ll address your other questions, and we hope the above sheds some clarifying light on many of them.
>
> *General Applicability* Our theorems follow from differential privacy of the random bits, but this can generally be obtained using standard methods designed to give differential privacy of the data points. For example, consider an algorithm in which each random bit affects the final output model only through its effect on a single data point: this is the case if e.g. random bits are used to select whether a given point appears in a minibatch or not, or if we have a SISA like model in which the points are partitioned rather than selected independently to appear in each shard. In this case, (standard) differential privacy of the data points implies differential privacy of the random bits and causes our theorems to apply. Alternatively, suppose (as in the SISA like model we focus on) that each string of random bits affects the output only through its effect on a single shard. In this case, any method that protects the privacy of the constituent models (like private vote aggregation, as we consider in this paper --- for which there are a number of techniques in the work we cite---, but also private model averaging) satisfies the conditions of our theorem.
>
> *Accuracy guarantees* Because we are interested in the non-convex setting, it is in general impossible to provide worst case theoretical accuracy guarantees (just as it is in the original SISA framework, which has no worst-case accuracy guarantees). However, we inherit the accuracy guarantees of [Papernot et al., 2018, Dwork and Feldman, 2018] who study private aggregation in general: namely, our predictions will be correct whenever (roughly) $k/2 + 1/\epsilon$ of the models we are aggregating over produce the correct prediction. Note that the constituent models do not need to be trained privately. Thus, whenever $k \gg 1/\epsilon$ and we have sufficient data such that training on $n/k$ datapoints can be expected to produce a model with high accuracy (using any training method), then we should also expect our private aggregation method to produce high accuracy predictions. This can be formalized into a worst-case theorem for convex models, but we did not do this because our focus on this paper is on non-convex models (there are existing deletion methods for training convex models that are already robust to adaptivity).
>
> *Unlearning Parameters* As noted above, for accuracy, we want that $k \geq 1/\epsilon$. As you correctly note in your review, for unlearning guarantees, we also want that $\epsilon \leq \alpha/\sqrt{k}$. We note that both of these conditions can be satisfied simultaneously when $k=1/ \epsilon$ and $\epsilon \leq \alpha^2$. If we think of $\alpha$ as a small constant, we can also think of $\epsilon$ as a (smaller) constant.
>
> *Runtime Guarantees and the original SISA* We do provide runtime guarantees in Thm. 4.2, but as we note at the outset, the main contribution of the paper is to provide the _first_ method with any data deletion guarantees beyond full retraining for non-convex models. In particular, SISA does not provide any such guarantees at all. The reviewer is correct that the run-time of SISA remains constant with a partition of the data --- but this is not the issue. As we show in Section 5, SISA fails at its main goal in the presence of an adaptive deletion sequence, which is to make retraining indistinguishable from deletion.
>
> *Experiments* As we note above, privacy of datapoints implies privacy of random bits when the random bits affect the model only through the influence of a single data point, as they do when the randomness is used to select minibatches in private SGD.
>
> *Ullah et al.* Thank you for pointing out this missing reference --- it was an oversight, and we will discuss it in the next version. However, we want to clarify that _the method from Ullah et al. does not give deletion guarantees for adaptive sequences_, nor does it claim to. (In fact they make no reference to it, since the issue of adaptivity seems to have gone unnoticed until our work. The definitions they give do not capture adaptive sequences). To see why, note that the argument in Ullah et al. relies on a coupling between the gradient descent trajectories with and without the deleted point included. This coupling is enabled by Gaussian noise added to the gradients. However, if the deleted points depend on the realized trajectories, (as they would if the deletions are selected adaptively), then the noise addition is no longer independent and unbiased conditional on the deletion sequence, and so invalidates the proposed coupling.  We stress that this does not contradict the claims in their paper, since they do not claim that they support adaptivity.

---

> > ### Comment · Reviewer_bBvQ · 2021-09-01
> > **Response**
> >
> > I thank the authors for the response. In particular, thanks for clarifying my misunderstanding. Based on this as well as other reviews, I will increase the score. Here are some final suggestions:
> >
> > 1. The authors claim that method also works for method which guarantee differential privacy of the data points, like standard noisy SGD. It would be instructive if the authors can instantiate their guarantees for this method in detail.
> >
> > 2. Accuracy guarantees: I still think that it would be helfpul if the authors give accuracy guarantees in some simple convex settings -- the authors already sketch part of the argument in the response, so it should not be much work. Also, mention what existing methods in convex settings are already robust to adaptive sequences (as the authors claim), and why does the same method not work in the non-convex case, which is the focus of this work.

---

> > > ### Author Response · Authors · 2021-09-01
> > > **Thanks**
> > >
> > > Thank you! We will take your suggestions to heart.
> > >
> > > Re: the convex methods that are robust to adaptivity --- we briefly sketch this in the related work, but here is an elaboration:
> > >
> > > As we show in our work, any method that is differentially private in the randomness used to prove the deletion guarantee satisfies adaptive deletion guarantees. A (trivial) special case of this is when the deletion algorithm is deterministic. A deterministic deletion method must lead to _exactly the same model_ that would have resulted from retraining, so is not in general possible for non-convex models, but is possible for linear regression, for which the update has a closed form. So algorithms like this are already robust to adaptive deletion sequences.
> > >
> > > Another special case is when the randomness on which the deletion guarantee relies is sampled freshly at each round, after the deletion request comes in. (Hence the deletion request cannot depend on the randomness, and the conditions of our theorem are satisfied). Deletion methods for strongly convex models take advantage of this by running a number of iterations of gradient descent such that the convergence theorem promises that the solution is within some $\alpha$-ball of the exact optimizer. It then suffices to add enough noise to the resulting parameters to obscure the difference between any two points that are within distance $2\alpha$ of one another. This technique crucially relies on being able to analytically prove that gradient descent converges to a unique global optimum, and so only applies to (strongly) convex models.

---

> > > > ### Author Response · Authors · 2021-09-01
> > > > **One more comment**
> > > >
> > > > We note that the fact that these previous convex methods are robust to adaptive data deletion sequences can be viewed as (an easy) corollary of our main theorem. Robustness to adaptive data deletion sequences was not stated in previous work, which had not formalized or considered adaptive deletion sequences.

---

### Official Review · Reviewer_cu9f · 2021-07-19

**Rating:** 6
**Confidence:** 4

**Summary:**

 considers the problem of machine unlearning against an adaptively generated sequence of unlearning requests. The authors work with the SISA framework (Bourtoule et. al.) for machine unlearning where the key idea is as follows: Divide the data into k shards, train separate models on each shard and then aggregate their prediction to get the final output on a test sample. During unlearning, figure out the shard that contains the point that wants to be deleted and then retrain the model on that shard and aggregate, keeping the rest of the models unchanged.

The goal of this unlearning framework is that the output model after unlearning is similar (in a probabilistic sense) to the model that we get after retrained from scratch on the left over data points, and at the same time ensure that the unlearning mechanism is computationally more efficient as compared to retraining from scratch. One benefit of the provided SISA mechanism is that it achieves both of these goals for arbitrary learning algorithms and for convex / non-convex models.

The main contributions of the paper are as follows:
(a) They show that the SISA algorithm from Bourtoule et. al. fails to provide good performance guaranteed when used for unlearning adaptively generated delete requests.

(b) They provide a general reduction using differential privacy to get unlearning algorithms that work for adaptive delete requests from unlearning algorithms that guarantee success with oblivious delete requests only. This reduction is based on using differential privacy to constraint the adversary from generative adaptive unlearning samples.

(c) They use this general framework to get a SISA algorithm for adaptive unlearning requests. They show experiments comparing their algorithm to standard-SISA.


**Limitations And Societal Impact:**

Yes

**Main Review:**

I found the paper somewhat unsatisfactory. My main reasons are:
(a) An important baseline algorithm to compare to is the one based on simple group DP. The algorithm is as follows:

Learning phase (L):
- Train a learning algorithm A^{single} on the entire dataset D.
- Make it (\eps, \delta)-group DP by adding noise to the output / while training, so that the learnt model is private w.r.t. datasets differing in m points.

Unlearning phase (UL):
- Do not do anything except for the following restarts.
- On every (m + 1)-th request, retrain the model from scratch on left over data points using the learning procedure L.

In the above, the unlearning phase simply does not do anything,  except for at every (m+1)-th request when it retrains from scratch on the left over datasets (and adds noise again to get (\eps, \delta) group DP w.r.t. m points). The group DP guarantees should ensure that we unlearn against adaptive adversaries.

Discussion of the above algorithm w.r.t. the algorithm provided in the paper:
- The algorithm in the paper also adds noise for the prediction after every step. Is there any difference in the scale of the noise that is added in the provided algorithm and the one needed to get group DP (above).
- As shown in Algorithm 4 in the appendix, the algorithm ini the paper also retrains from scratch after a certain privacy threshold is attained.
- Under observation 1 and 2, the algorithm provided in the paper seems to be very similar to the one I described above. What are the differences? I think it is important to verify, at least experimentally, whether the provided algorithm is better than the above simple DP based algorithm in terms of performance.
- The above mentioned algorithm does nothing during unlearning and is thus computationally better than the proposed algorithm in the paper which retrains on the relevant shared after every delete requests.

(b) There is a small issue with the illustrative example given in Section 4.1. Here, the behavior of the model output by retraining algorithm is slightly different that what was stated. According to the setup, every data point has exactly two copies. Once we delete the samples that are in different shards, we are left with at most one copy of those data points in the dataset. Thus, no matter how we split the dataset during retraining, the aggregating algorithm will fail to perform well on these points. Thus the performance of the retraining from scratch is smaller than ⅔ (as stated in the paper).

Furthermore, it seems that in order to show a meaningful separation, the unlearning algorithm needs to delete a constant fraction of the dataset. If the unlearning requests leave spare any points which were in different shards in the first split, then with constant probability they can go to the same shard in the second split. This would further diminish the gap between the performance of the unlearnt model and the model that retrains from scratch. I should also mention that it has been observed in some other works (e.g. Theorem 2 in Sekhari et. al. 2021): That paper seems to show that there exists some problems where it is impossible to delete a constant fraction of the dataset while still retaining test performance. Is there any deeper connection between the two examples that would suggest limits on unlearning?

Can we show that there are problems where the model after unlearning and retraining from scratch are significantly different after unlearning m points adaptively, where m = o(n) (little O).

(c) Definition 2.3 requires the base algorithm A to be randomized. This seems unnecessary - why add any noise to the base algorithm when there are no unlearning requests yet. The added noise can hurt the test performance.

(d) As shown in the experiments, the performance for even simple problems like CIFAR-10 and MNIST degrade for the provided algorithm (0.5-0.6 for CIFAR 10 and 0.9 for MNIST). Is that necessary? What would be the performance of the algorithm that just adds noise to ensure DP (or group DP) and then does nothing for unlearning?

I think that it is important to compare the performance of the provided model against vanilla DP to say that the unlearning algorithm is doing anything better. As of now, it seems that in terms of both final accuracy and running time, group DP based simple approaches may be better.

Paper writing:
It is a well written paper. In order to improve clarity, it might be useful to make explicit what the different functions and operators are for the illustrated examples.


I would be happy to change my score depending on the answers to my questions above.


**Time Spent Reviewing:**

10-14 hours

---

> ### Author Response · Authors · 2021-08-10
> **Response**
>
> Thank you very much for reading our paper so closely and the obvious effort you put in providing insightful feedback. It will improve the paper. Here we respond to your questions and comments:
>
> *The static differential privacy baseline* You are right that this is a natural baseline, and in the next revision of the paper we will discuss it: *we improve over it substantially*. Briefly (for the benefit of other reviewers and area chairs), the baseline proposed by Reviewer cu9f depends on the following observations: A training algorithm that satisfies $\alpha$-DP also satisfies an $\alpha$-deletion guarantee for _a single deletion_ when the deletion algorithm simply does nothing, and differential privacy has the following “group privacy” guarantee: an $\epsilon$-DP algorithm is also $m\epsilon$-DP for groups of size m. Thus given a parameter m, a baseline algorithm trains a model subject to $\alpha/m$-DP, and then fully retrains every m deletions (doing nothing on other rounds). Briefly, we improve over this baseline because for us, privacy degrades due to _composition_ rather than _group privacy_. Under composition, differential privacy degrades as $\sqrt{m}\epsilon$ rather than as $m\epsilon$. Thus at the same privacy parameter $\alpha/m$, for any constant $k$, our method requires full retraining roughly every $O(m^2)$ rounds, rather than every $m$ rounds required of the baseline. For reasonable values of $k$, this gives a substantial computational improvement.
>
> *Our theoretical example in Section 5.1* Our calculation is correct, but your comment alerts us to a possible source of confusion in the exposition. Recall that there are two copies of each training point. We delete points that the ensemble classifies correctly: note that because the ensemble classifies points deterministically (and therefore gives the same classification to identical points) for any point that is classified correctly, we delete _both_ copies. Thus there remain two copies of any point that was not deleted, which is why the expected retraining error remains 2/3. We will add a sentence to clarify.
>
> You are right that our examples in Section 5 (both the theoretical one and the empirical one) involve large numbers of deletions to witness large gaps in the accuracy of the resulting models. But this is just for expositional clarity in section 5.1 and experimental efficiency in 5.2. Note that to falsify deletion guarantees, we do not need large gaps in accuracy --- we only need large gaps in the likelihood of an observed event. A Chernoff bound implies that we can obtain high probability separations between the empirical mean of a bounded random variable from $n$ samples whenever their true means differs by at least $\sim 1/\sqrt{n}$. _This directly implies that our theoretical construction in Section 5.1 continues to falsify deletion guarantees of SISA even when the fraction of points we delete is only $O(1/\sqrt{n})$, rather than $\Omega(1)$._
>
> *Definition 2.3* It is important that the training algorithm $A$ be randomized (i.e. the method that produces the model we release even before deletion requests have come in) because the notion of deletion we and past works use asks for approximate statistical indistinguishability between the model resulting from a deletion sequence and the model resulting from retraining. If the model resulting from retraining was a deterministic function of the remaining data, this would constrain the deletion algorithm to be (almost) deterministic as well (and to almost always lead to the same outcome). Adding randomization adds flexibility to the design space, but the randomization must be added to both training and deletion algorithms if they are to result in similar distributions over outcomes.

---

> > ### Comment · Reviewer_cu9f · 2021-09-01
> > **Thanks for your response!**
> >
> > Thank you! This does clarify some of my concerns.
> >
> > I am not yet convinced why your privacy degrades due to composition instead of group privacy, and would really like to see empirical / formal theoretical justifications to compare with just running group-DP.
> >
> > My other concern is that I am not on the same page in terms of what we want to compare to. Suppose there were no unlearning requests. Still the output of your learning algorithm is randomized. The model thus looses some performance due to added noise. It seems wrong to add noise even when there were no unlearning requests. Do you agree ?
> >
> > Sorry for the a late response!

---

> > > ### Author Response · Authors · 2021-09-01
> > > **Thanks for the reply!**
> > >
> > > Thanks for writing back! To answer your questions:
> > >
> > > 1) Composition vs group privacy: Note that the hypothesis to our main theorem (Thm 3.1) is that the sequence of publishing functions $f_1,\ldots,f_t$ together satisfy $(\epsilon,\delta)$-differential privacy. The privacy of a sequence of individually differentially private algorithms follows from adaptive composition. Roughly speaking (ignoring constants), if each of $f_1,\ldots,f_t$ satisfy $\epsilon$-DP for $\epsilon \ll 1/\sqrt{t}$, then modern composition theorems state that the sequence satisfies $(O(\sqrt{t \log(1/\delta)}),\delta)$-DP. Note that the $\epsilon$ parameter here degrades only with the square root of $t$, which is where the gain over the static DP baseline comes from, which relies on the group privacy property of differential privacy, which scales linearly with the size of the group.
> > >
> > > 2) Randomizing the initial model: It is counter-intuitive, but nevertheless true that to obtain deletion guarantees, it is always necessary to randomize the initial model if the deletion algorithm involves any randomness at all, even though this is wasteful in retrospect in the event that there turn out not to be any deletion requests. To see why this is, note that an approximate deletion guarantee requires that the distribution over models that results from applying the deletion algorithm be close to the distribution over models that results from freshly applying the initial training algorithm. If the initial algorithm is deterministic, then the distribution that results from retraining is a point mass, and so in order to match this, the deletion algorithm must also be deterministic (and result in exactly the same model as full retraining). This is generally not possible except in the simplest of settings (e.g.. linear regression).
> > >
> > > Please let us know if you have further questions, we are happy/eager to provide answers!

---

### Decision · Program_Chairs · 2021-09-28

**Decision:**

Accept (Poster)

**Comment:**

This paper identifies a new concern in machine unlearning, the role of adaptivity in the request sequence. Specifically, if removal requests may be adaptive, then the unlearning guarantees may be violated. The authors also give a differential privacy based method to mitigate this issue. This is an interesting new phenomenon, and the paper should be accepted.

The authors are suggested to pay attention to the presentation comments made by the reviewers: as machine unlearning is a relatively new field, it is important to make the early papers as well written as possible.

(As one minor personal comment, I disagree with the authors' response that a drop from 97% to 91% accuracy (MNIST k=6) is relatively small, I would say this is significant when the error rate is so small.)

**Consistency Experiment:**

NeurIPS has a long history of experimentation. In 2014, NeurIPS ran an experiment in which 10% of submissions were reviewed by two independent committees to quantify the randomness in the review process. This year, we repeated a variant of this experiment to see how the quality of the review process has changed over time.  This paper was part of the experiment and was therefore assigned to two committees (consisting of reviewers, an Area Chair, and a Senior Area Chair) that reached independent decisions.  If both committees made the same recommendation, this recommendation was followed. If a single committee recommended acceptance, the paper was accepted (with the exception of a few cases in which the other committee identified what we considered a fatal flaw, e.g., an error in a key result).

Both committees reached the same decision: **Accept (Poster)**

The other committee assigned to the paper recommended **Accept (Poster)**.  You can find the other set of reviews, along with any follow up discussion with the authors here:
https://openreview.net/forum?id=nWSZ30wrEw3